# Steroid-dependent switch of OvoL/Shavenbaby controls self-renewal versus differentiation of intestinal stem cells

Sandy Al Hayek[1,2,3] , Ahmad Alsawadi[3], Zakaria Kambris[4] , Jean-Philippe Boquete[5], Jérôme Bohère[3] , Clément Immarigeon[3], Brice Ronsin[3], Serge Plaza[3,†], Bruno Lemaitre[5] , François Payre[3,*] & Dani Osman[1,2,**]

## Abstract

Adult stem cells must continuously fine-tune their behavior to regenerate damaged organs and avoid tumors. While several signaling pathways are well known to regulate somatic stem cells, the underlying mechanisms remain largely unexplored. Here, we demonstrate a cell-intrinsic role for the OvoL family transcription factor, Shavenbaby (Svb), in balancing self-renewal and differentiation of *Drosophila* intestinal stem cells. We find that *svb* is a downstream target of Wnt and EGFR pathways, mediating their activity for stem cell survival and proliferation. This requires post-translational processing of Svb into a transcriptional activator, whose upregulation induces tumor-like stem cell hyperproliferation. In contrast, the unprocessed form of Svb acts as a repressor that imposes differentiation into enterocytes, and suppresses tumors induced by altered signaling. We show that the switch between Svb repressor and activator is triggered in response to systemic steroid hormone, which is produced by ovaries. Therefore, the Svb axis allows intrinsic integration of local signaling cues and inter-organ communication to adjust stem cell proliferation *versus* differentiation, suggesting a broad role of OvoL/Svb in adult and cancer stem cells.

**Keywords** *Drosophila*; enterocyte differentiation; intestinal stem cells; OvoL transcription factors; Wnt and EFGR pathways
**Subject Categories** Cancer; Signal Transduction; Stem Cells & Regenerative Medicine
**The EMBO Journal (2021) 40: e104347**

## Introduction

Living organisms are constantly exposed to aging and environmental challenges that disturb cell functions and ultimately lead to cell death. To maintain homeostasis, most adult organs are regenerated by self-renewing stem cells, which differentiate to replace dead cells and replenish damaged tissues. The highly regenerative digestive system is kept intact during adulthood by the activity of resident intestinal stem cells. *Drosophila* intestinal stem cells have emerged as a powerful system to understand the signaling networks underlying stem cell biology and their implication in cancers (reviewed in (Li & Jasper, 2016; Perochon *et al*, 2018)).

The adult fly intestine consists of a compartmentalized epithelium (Buchon *et al*, 2013), which shares anatomical and physiological similarities with its mammalian counterpart. *Drosophila* intestinal stem cells (ISCs) are small diploid cells scattered along the basement membrane (Micchelli & Perrimon, 2006; Ohlstein & Spradling, 2006). In steady-state conditions, ISC divide asymmetrically to generate a new stem cell and a transient post-mitotic progenitor cell called enteroblast (EB) (Ohlstein & Spradling, 2007). ISCs and early EBs express Escargot (Esg), a transcription factor of the Snail/Slug family that maintains diploidy and prevents premature differentiation (Korzelius *et al*, 2014; Loza-Coll *et al*, 2014). EBs progressively acquire characteristics of polyploid absorptive enterocytes (ECs), representing the main population of intestinal cells (Ohlstein & Spradling, 2007). The second type of differentiated intestinal cells is hormone-secreting enteroendocrine cells (EEs). They emerge from a separate pool of progenitors (Biteau & Jasper, 2014; Zeng & Hou, 2015), called pre-enteroendocrines (pre-EEs), which express markers of both ISCs (Esg) and EEs (Prospero).

The evolutionarily conserved Notch pathway establishes the asymmetry between ISCs and EBs (Micchelli & Perrimon, 2006;

---

1   Faculty of Sciences III, Lebanese University, Tripoli, Lebanon
2   Azm Center for Research in Biotechnology and its Applications, LBA3B, EDST, Lebanese University, Tripoli, Lebanon
3   Centre de Biologie du Développement (CBD), Centre de Biologie Intégrative (CBI), Université de Toulouse, CNRS, Toulouse, France
4   Biology Department, Faculty of Arts and Sciences, American University of Beirut, Beirut, Lebanon
5   Global Health Institute, School of Life Sciences, Lausanne, Switzerland
    *Corresponding author (Lead contact). Tel: +33 561 556 348; E-mail: francois.payre@univ-tlse3.fr
    **Corresponding author. Tel: +961 718 880 65; E-mail: dani.osman@ul.edu.lb
    †Present address: Laboratoire de Recherche en Sciences Végétales (LSRV), CNRS, UPS, Castanet-Tolosan, France

---

Ohlstein & Spradling, 2007; Bardin *et al*, 2010; Perdigoto *et al*, 2011). ISCs express Delta, a ligand that activates the Notch receptor in daughter EBs, as seen by Su(H) expression. The EC fate requires high levels of Notch, whereas lower Notch activity induces the production of EEs that maintain Prospero expression. Gut homeostasis relies on a tight regulation of ISC division through cooperative activity of conserved developmental signaling pathways, such as the epidermal growth factor receptor (EGFR), Wnt, and JAK/STAT pathways (Jiang & Edgar, 2009; Biteau & Jasper, 2011; Jiang *et al*, 2011). Despite the wealth of knowledge accumulated on the role of signaling pathways in regulating ISC maintenance, division, and differentiation, the intrinsic mechanisms by which ISCs integrate these cues remain largely unknown.

During embryogenesis, the activity of Wnt and EGFR pathways in the epidermis is mediated by a common target gene, *ovo/shavenbaby* (*svb*), which encodes a transcription factor governing epidermal differentiation (Payre *et al*, 1999). The Svb factor undergoes post-translational processing from a repressor (Svb$^{REP}$) to an activator (Svb$^{ACT}$) via limited proteasome degradation (Zanet *et al*, 2015). Svb maturation is triggered by Polished rice (Pri) peptides (Kondo *et al*, 2010), which are founding members of a growing family of peptides translated from small open reading frames, called smORF peptides (Saghatelian & Couso, 2015; Plaza *et al*, 2017). *ovo/svb* is also critical for maintenance and differentiation of the germline (Mevel-Ninio *et al*, 1995). There are two Svb germline-specific isoforms, called OvoA and OvoB, which are insensitive to Pri peptides (Kondo *et al*, 2010) and act as constitutive repressor and activator (Andrews *et al*, 2000), respectively. Throughout development, the production of Svb$^{ACT}$ in somatic tissues is triggered by periodic peaks of ecdysone, the main steroid hormone in insects. Upon hormone binding, the ecdysone receptor (EcR) directly activates the expression of *pri,* triggering, in turn, Svb processing (Chanut-Delalande *et al*, 2014). The ecdysone signaling pathway has also wide-ranging functions in adults, including regulation of stress resistance, nutritional state, and reproduction (Uryu *et al*, 2015).

Ovo/Svb defines a metazoan-specific family of transcription factors, comprising three paralogs in vertebrates called *OvoL1-3*, which are crucial regulators of epithelial lineage determination and differentiation. For example, human OvoL2 is required for the maintenance of corneal epithelium cells (Kitazawa *et al*, 2016) and its alteration is a major cause of inherited corneal dystrophies (Davidson *et al*, 2016). OvoL factors have been involved in the metastatic/stemness potential of various tumors, including in breast (Roca *et al*, 2013), prostate (Fu *et al*, 2016), lung (Wang *et al*, 2017), and colorectal (Ye *et al*, 2016) cancers. Moreover, OvoLs also act for the repair of epithelial tissues from stem/progenitor cells, *e.g.*, for epidermal and mammary regeneration (Watanabe *et al*, 2014; Haensel *et al*, 2019). In the flatworm, OvoL/Svb is expressed in eye progenitors and required for eye regeneration from multipotent stem cells (Lapan & Reddien, 2012). Hence, a growing body of evidence suggests a role of OvoL/Svb in stem/progenitor cells across animals. Indeed, we recently found that Svb is required for the survival of renal nephric stem cells (RNSCs) in adult flies, via direct interaction with Yorkie (a.k.a. YAP/TAZ), the nuclear effector of the Hippo pathway (Bohere *et al*, 2018). RNSCs derive from progenitors that also produce intestinal stem cells (Xu *et al*, 2018), suggesting a broader function of Svb in adult stem cells.

Here, we demonstrate that the Shavenbaby transcription factor is essential to adult midgut homeostasis. Importantly, proteasome-mediated processing allows Svb isoforms to exert antagonistic functions along the ISC lineage. Through clonal analysis of a null allele of *svb*, and cell type-specific RNAi knockdown or overexpression, we conclude that the processed Svb$^{ACT}$ is required to maintain ISCs and sufficient to induce their self-renewal. In contrast, the unprocessed Svb$^{REP}$ directs differentiation into ECs, in which it is further required to maintain the differentiated state. *svb* expression in either ISC/EBs, or ECs, is driven by separate regulatory networks. Results from a large *in vivo* screen reveal that *svb* enhancers are directly regulated on the one hand by Wnt and EGFR local signaling for ISC/EB survival and self-renewal, and, on the other hand, by intrinsic regulatory factor Pdm1 for EC differentiation. Moreover, recent studies show that the systemic steroid hormone ecdysone, which is produced in ovaries (Uryu *et al*, 2015), increases proliferation and regulates the fate of stem cells in the intestine (Ahmed *et al*, 2020; Zipper *et al*, 2020). Our data suggest that these effects of ecdysone are due, at least in part, to the activation of *pri* expression that triggers, in turn, Svb processing. Together, these results reveal the dual role of OvoL/Shavenbaby in stemness versus differentiation and provide a first molecular frame to explain how local and systemic regulatory signals, in coordination with intrinsic cues, are integrated within the adult stem cell lineage.

# Results

### Svb is required to maintain adult intestinal progenitors

*svb* expression is driven by a large array of enhancers, which collectively define at single-cell resolution the pattern of epidermal differentiation in the embryo (Sucena *et al*, 2003; McGregor *et al*, 2007; Frankel *et al*, 2011; Preger-Ben Noon *et al*, 2016). To monitor *svb* expression in the adult midgut epithelium, we tested the activity of main *svb* enhancers. While one *svb* enhancer (*7*) was active in terminally differentiated cells (see below), we found that the *E* enhancer (Fig 1A) drives specific expression in *esg*$^+$ progenitors (Fig 1B), *i.e.*, in stem cells (ISCs) and enteroblasts (EBs). Dissection of the *E* enhancer (5kb) delineated two separate elements called *E3N* (292 bp) and *E6* (1kb) that each drives similar expression in intestinal progenitors (Figs 1C and EV1A).

To investigate the function of Svb in adult intestinal stem cells, we used targeted RNAi depletion using conditional and temperature-sensitive drivers, typically induced in 3-day-old mated females. As a first step, we used the *esg*$^{ts}$ driver (Micchelli & Perrimon, 2006) to drive *svb* knockdown in adult progenitor cells (*esg*$^+$). Knockdown of *svb* in the *esg*$^+$ population for 2 weeks led to almost complete disappearance of ISCs, as seen by loss of *Delta-lacZ*$^+$ cells (Fig 1D), as well as loss of EBs marked by *Su(H)-GBE-lacZ* (Fig 1D′). Consistently, *svb* depletion specifically targeted either in stem cells by the *ISC*$^{ts}$ system (Wang *et al*, 2014), or in enteroblasts by using *Su(H)*$^{ts}$ (Zeng *et al*, 2010), caused the loss of ISCs or EBs, respectively (Figs 1E and EV1B). In contrast, *svb* knockdown did not affect the enteroendocrine lineage (Fig EV1C and D). Hence, these data show that Svb is specifically required for the maintenance of ISCs and EBs.

The loss of stem/progenitor cells upon *svb* knockdown could be due to premature differentiation and/or cell death; we then performed a series of genetic experiments to discriminate between these possibilities. The *act^{ts}F/O* system allowed random knockdown of *svb* in dividing intestinal cells and their progeny (marked by GFP), leading to a strong decrease in both the number and size of

GFP⁺ clones (Fig EV1E). We next used the mosaic analysis with a repressible cell marker (MARCM) technique (Lee & Luo, 2001) to generate positively marked clones (GFP⁺) in the midgut epithelium for a null mutation in *svb* (Delon *et al*, 2003). *Svb*-mutant clones were rare and far smaller than control clones, being often restricted to single cells (Fig 1F). Therefore, the loss of stem cells observed

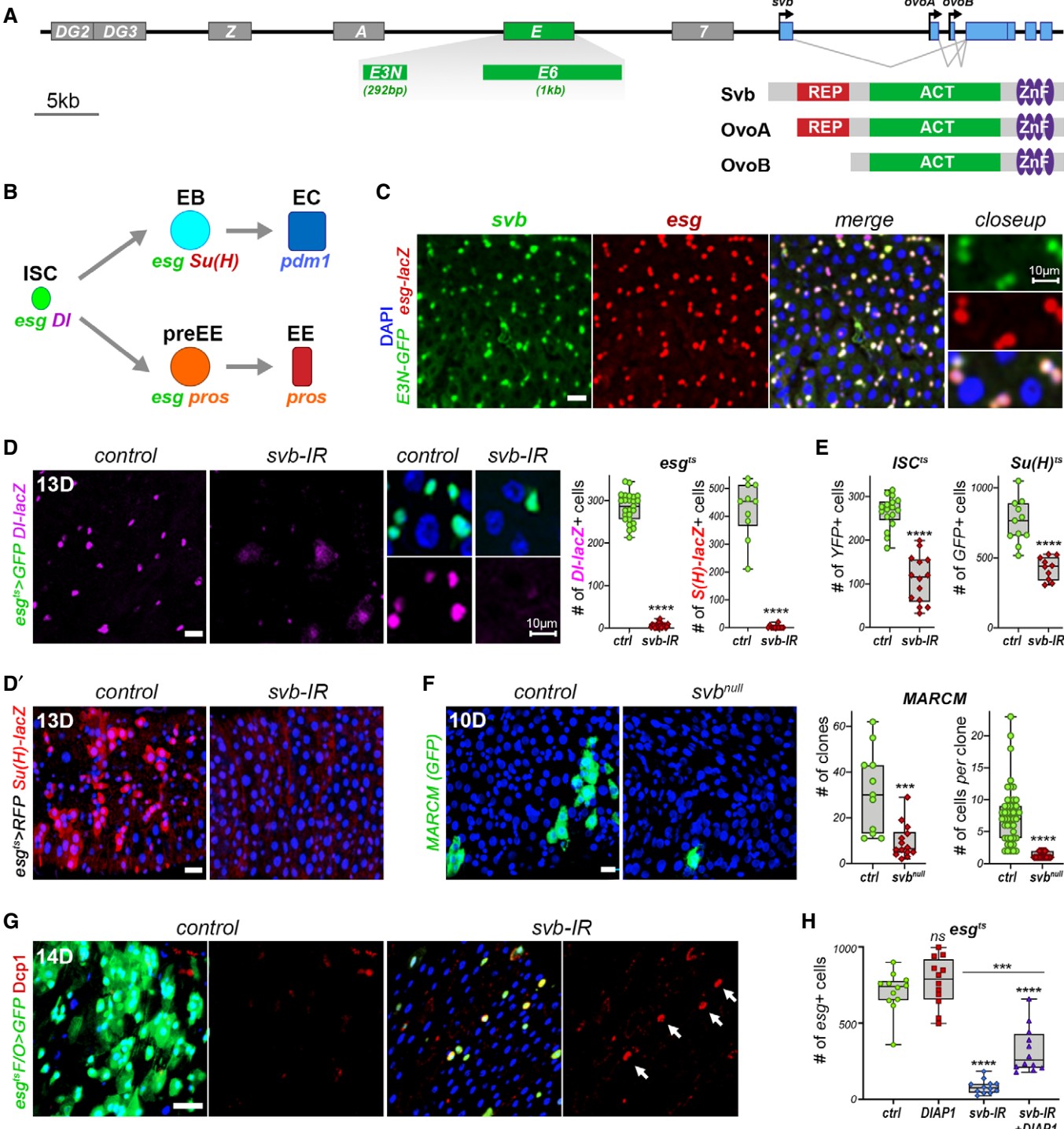

**Figure 1.**

**Figure 1.** *sub* is expressed in ISC/EBs and is required for their maintenance.

A    Schematic representation of the *sub* locus, showing location of enhancers as well as functional organization of somatic (Svb) and germline (OvoA, OvoB) protein isoforms. Red and green boxes represent the repressor and activator domains, respectively; purple ovals depict the DNA-binding zinc fingers.

B    The adult intestinal stem cell lineage, with markers of stem cells (Dl), enteroblasts (Su(H)) and enterocytes (Pdm1). Esg is expressed in progenitor cells gathering stem cells (ISC), enteroblasts (EB), and pre-enteroendocrines (preEE). Both pre-EEs and mature enteroendocrine cells (EE) express Prospero (pros).

C    Posterior midgut showing expression of the *E3N sub* enhancer (GFP, green) in ISC/EBs, as shown by co-staining with *esg-lacZ* (β-Gal, red).

D, D′    Staining for *Dl-lacZ* (purple) or *Su(H)-lacZ* (red) in *esg^{ts}* midguts expressing GFP alone (control), or expressing *sub*-RNAi. Samples were stained for GFP (green) and β-Gal. Close-ups show separate channels for GFP and β-Gal. The graphs show quantification of the number of ICS *(Dl-positive)* and EBs (*Su(H)-positive*).

E    Quantification of the number of YFP-positive cells (left) and GFP-positive cells (right) in control and upon expression of *sub*-RNAi driven by *ISC^{ts}* and *Su(H)^{ts}*, respectively.

F    Posterior midguts containing control or *sub^{R9}* MARCM clones (GFP, green), and quantification of the number of clones, and of the average number of cells *per* clone.

G    *esg^{ts}F/O* midguts expressing GFP alone (control) or expressing *sub*-RNAi. Samples were stained for GFP (green) and the apoptotic marker cleaved Dcp1 (red). Arrows highlight GFP-positive cells that are also positive for Dcp1.

H    Quantification of GFP-positive cells *per* posterior midgut in *esg^{ts}* expressing GFP alone (ctrl), or expressing DIAP1, *sub*-RNAi, and *sub*-RNAi⁺ DIAP1.

Data information: Boxes extend from the $25^{th}$ to $75^{th}$ percentiles, whiskers from min to max, the horizontal line in each box is plotted at the median; data were collected from three independent replicates. *P* values from Mann–Whitney tests (D,E,F) and one-way ANOVA (H) are ns > 0.05, * < 0.05, ** < 0.01, *** < 0.001, **** < 0.0001. DAPI is blue, scale bars, 20 μm, except in close-ups (10 μm).

Source data are available online for this figure.

upon *svb* inactivation was likely resulting from their death, a conclusion we further tested by lineage tracing experiments. We used the repressible dual differential stability markers (ReDDM) approach (Antonello *et al*, 2015) in which *esg*⁺ cells express both short (mCD8::GFP) and long (Histone::RFP) half-lives proteins, the latter persisting in differentiated progeny (GFP negative) for several weeks (Fig EV1F). ReDDM results confirmed that ISC/EBs did not prematurely differentiate upon *svb* knockdown, since the loss of progenitors (GFP⁺/RFP⁺) was not paralleled by an increased number of differentiated cells (GFP⁻/RFP⁺). We also generated clones of intestinal cells using the *esg^{ts}F/O* system (Jiang *et al*, 2009), which marks both ISC/EBs and their descendant progeny by GFP, and stained for the apoptotic marker cleaved-Dcp1. Two weeks after induction, large GFP⁺ clones and only rare apoptotic cells were observed in control midguts. In contrast, *svb* knockdown led to sparse GFP⁺ cells, often positive for Dcp1, thus demonstrating that progenitors lacking *svb* underwent apoptosis (Fig 1G). Accordingly, expression of the apoptosis inhibitor DIAP1 was sufficient to significantly rescue the ISC/EB population following *svb* knockdown (Fig 1H).

Hence, loss of *svb* leads to a loss of stem/progenitor cell population, demonstrating that Svb is required for their maintenance and protection from apoptosis.

### The Pri/Ubr3/proteasome axis controls Svb function in stem cells

Svb is translated as a large (1,354 aa) repressor (Svb^REP) that is processed into a shorter (910 aa) transcriptional activator (Svb^ACT) (Kondo *et al*, 2010). This switch is gated by Pri peptides that bind to and activate the E3 ubiquitin ligase Ubr3, triggering Ubr3 binding to Svb (Zanet *et al*, 2015). The Ubr3/UbcD6 complex then ubiquitinates Svb, inducing in turn Svb processing via limited proteasome degradation of its N-term repressor domain (see Fig 2C). Originally identified in the epidermis (Kondo *et al*, 2010; Chanut-Delalande *et al*, 2014), there is growing evidence that Pri-dependent processing underlies Svb function in other somatic tissues (Pueyo & Couso, 2011; Ray *et al*, 2019), including in adult stem cells (Bohere *et al*, 2018).

To investigate whether Svb processing regulated stem cell fate, we first examined *pri* expression in the adult midgut. Profiling of reporter lines covering the entire *pri* locus (Chanut-Delalande *et al*, 2014) showed that three *pri* enhancers (*priA, priH,* and *priJ*) were active in ISC/EBs (Fig 2A and B). We also monitored a *Gal4* gene trap within *pri* gene that faithfully reflects the pattern of *pri* in many tissues (Galindo *et al*, 2007). This experiment confirmed *pri* expression in ISC/EBs, and not in large polyploid ECs (Fig 2B). Since *pri* was specifically expressed in stem/progenitor cells, we investigated its putative function by targeted knockdown. Upon 2 days of *pri*-RNAi induction in *esg*⁺ progenitors, the majority of GFP⁺ cells had disappeared from the midgut (Fig 2D). We also observed an acute loss of stem cells when *pri*-RNAi was driven using *ISC^{ts}* (Figs 2D and EV2A). Hence, loss of *pri* leads to a loss of stem/progenitor cells, demonstrating that, like *svb*, *pri* is required for their maintenance.

Throughout development, ecdysone signaling times *pri* expression through direct activation by EcR (Chanut-Delalande *et al*, 2014). We reasoned that if this hormonal control of *pri* expression was occurring in the adult midgut, cell-autonomous disruption of the ecdysone pathway should affect the behavior of ISCs. Consistent with this prediction, EcR knockdown—using two non-overlapping RNAi driven by *ISC^{ts}* or *esg^{ts}*—led to a loss of ISCs and EBs (Figs 2E and EV2B). Similar results were obtained when driving EcR-DN, a dominant negative form of the receptor, confirming that ecdysone signaling is required within ISC/EBs (Figs 2F and EV2C). Furthermore, expression of *pri* was able to rescue the loss of ISC/EBs caused by EcR-DN (Figs 2F and EV2C). These data indicate that ecdysone signaling is required for ISC homeostasis and that *pri* is a main target of EcR within adult stem cells.

Our observations supported a model in which Pri peptides act in ISC/EBs to trigger Ubr3-mediated processing of Svb. To test this model, we generated MARCM clones of intestinal cell homozygous mutant for a null allele of *Ubr3* (Zanet *et al*, 2015). As observed for *svb* mutants, clones lacking *Ubr3* were very rare and consisted of only a few cells (Fig 2G). Knockdown of *Ubr3* in

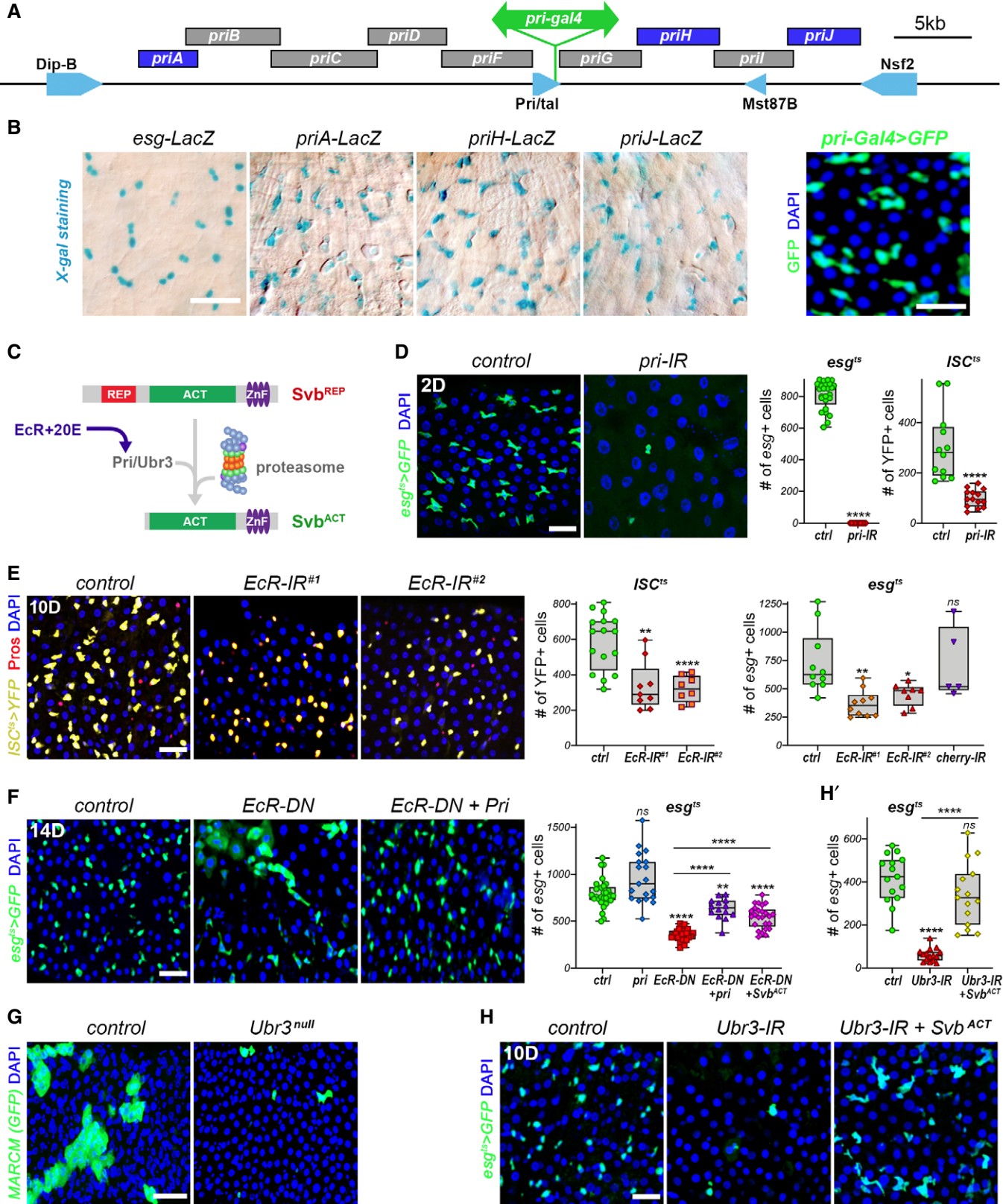

**Figure 2.**

**Figure 2. Pri/proteasome processing of Svb is required for ISC/EB maintenance.**

A   Schematic representation of the *pri* locus, with tested enhancers and location of the *pri-Gal4* gene trap insertion.
B   Expression of *priA*, *priJ*, or *priH* enhancers in the posterior midgut as seen by *lacZ* reporters (X-Gal staining, blue), and *pri-Gal4* gene trap expressing GFP (green).
C   Schematic representation of Svb maturation by proteasome processing, which is triggered by EcR-mediated expression of *pri*.
D   *esg^{ts}* midguts expressing GFP alone (control), or expressing *pri*-RNAi, and quantification of GFP-positive cells (green). The graph also shows the number of YFP-positive cells in *ISC^{ts}* midguts expressing YFP alone (ctrl), or *pri*-RNAi (see Fig EV2A).
E   *ISC^{ts}* midguts expressing YFP alone (control), or expressing two non-overlapping *EcR*-RNAi, and quantification of the number of YFP-positive cells (yellow). The graph also plots the number of GFP-positive cells in *esg^{ts}* midgut expressing GFP alone (ctrl), or expressing *EcR*-RNAi#1, *EcR*-RNAi#2, and *mcherry*-RNAi as an additional negative control (see Fig EV2B).
F   *esg^{ts}* midguts expressing GFP alone (control), or expressing EcR-DN, and EcR-DN^{+} *pri*. Samples were stained for GFP (green). The graph shows quantification of the number of GFP-positive cells in the different genotypes.
G   Posterior midguts containing control and *Ubr3* null MARCM clones (GFP, green).
H   *esg^{ts}* midguts expressing GFP alone (control), or expressing *Ubr3*-RNAi, and *Ubr3*-RNAi+ OvoB. Samples were stained for GFP (green). (H′) quantification of GFP-positive cells from H.

Data information: Boxes extend from the 25th to 75th percentiles, whiskers from min to max, the horizontal line in each box is plotted at the median; data were collected from three independent replicates. *P* values from Mann–Whitney tests (D) and one-way ANOVA (E,F,H′) are: ns:> 0.5, * < 0.05, ** < 0.01, **** < 0.0001. Blue is DAPI, scale bars, 20 μm.

progenitors (*esg^{ts}* > *Ubr3*-RNAi), or specifically in stem cells (*ISC^{ts}* > *Ubr3*-RNAi), also strongly decreased their number (Figs 2H and H′ and EV2D). If this loss of stem/progenitor cells was due to impaired Svb processing, then expression of a constitutive activator form of Svb (Andrews *et al*, 2000; Kondo *et al*, 2010) should suppress this phenotype. As predicted, co-expression of constitutive Svb^{ACT} with *Ubr3* RNAi significantly restored the pool of ISCs and EBs (Figs 2H and EV2D). Similar results were obtained by expressing both Svb^{ACT} and EcR-DN with the *esg^{ts}* driver, showing that Svb^{ACT} is also able to override *pri* downregulation (Fig 2F).

Taken together, these results demonstrate that ecdysone signaling is required within intestinal stem cells, in which it promotes the expression of *pri*. They further show that a main function of Pri and Ubr3 in the adult intestine is to trigger Svb maturation in order to maintain the pool of intestinal stem cells.

**Svb activator promotes stem cell renewal and sustains stemness**

Having shown that Svb^{ACT} is required to maintain stem/progenitor cells, we then asked whether elevated Svb^{ACT} activity could be sufficient to trigger stem cell hyperplasia in homeostatic conditions.

We used different means to increase Svb^{ACT} levels within ISC/EBs, *i.e.*, expression of the constitutively active form (Kondo *et al*, 2010), co-expression of Svb and Pri, or expression of a construct engineered to express the precise protein form normally resulting from Svb maturation (Ray *et al*, 2019). In all cases, we observed very similar results, with a strong increase in stem/progenitor population (Figs 3A and EV3A and B). For the sake of simplicity, the term Svb^{ACT} will be used in the following to collectively refer to these conditions. Examination of ISCs marked with *Dl-lacZ* showed that stem cells reached up to fourfold the normal population upon 2 weeks of *esg^{ts}*-driven Svb^{ACT} expression (Fig 3A). Similar increase in ISC population was also overserved when Svb^{ACT} was specifically targeted in ISCs (Fig EV3C). The expansion of stem cells resulted from over-proliferation, as seen by increased number of mitotic cells marked by phosphorylated-histone3 (PH3), while *svb* knockdown conversely reduced the mitotic index (Fig 3B). Taken together, these results thus demonstrate that high Svb^{ACT} is sufficient to trigger stem cell hyperproliferation.

Epithelial to Mesenchymal Transition (EMT) is a key process for the acquisition of stemness for both normal and cancer stem cells, and OvoL emerge as epithelial stabilizing factors able to counteract EMT (Nieto *et al*, 2016). Besides overgrowth, we then investigated whether Svb could as well influence epithelial features. ISCs are characterized by prominent basolateral accumulation of β-catenin (Ohlstein & Spradling, 2006), whereas β-catenin is restricted to apical cell junctions in differentiated cells. Like wild-type ISCs, Svb^{ACT} cell clusters displayed basolateral accumulation of β-catenin (Fig 3C). The same was also true for DE-Cadherin that is a hallmark of epithelial tissues (Nieto *et al*, 2016). In contrast, Scribble, a tumor suppressor that defines lateral domains, was reduced in both wild-type ISCs and Svb^{ACT} clones (Fig 3C). Previous work has shown that EMT-inducing factors Esg (Snail in mammals) and ZFh1 (Zeb1,2) are expressed in ISCs and required to maintain stemness and suppress differentiation (Korzelius *et al*, 2014). As in mammals, *miR8 (miR200)* downregulates Esg and Zfh1 levels in the fly midgut and *miR8* upregulation (*esg^{ts}* > *miR8*) induces precocious differentiation, resulting to the loss of stem cells (Antonello *et al*, 2015). We found that Svb^{ACT} was sufficient to overcome downregulation of EMT factors (*egs^{ts}* > *miR8^{+}* Svb^{ACT}), restoring the population of stem cells, *i.e.*, *esg*-GFP^{+} cells with enriched β-catenin in basolateral domains (Fig 3D). Notch promotes EMT and constitutive activation of Notch signaling in ISCs (*esg^{ts}* > NICD) enforces differentiation resulting in giant cells, with polyploid nuclei. Co-expression of Svb^{ACT} with NICD in ISC/EBs largely suppressed these defects (Fig 3E), restoring *esg^{+}* cells with normal-looking morphology and nuclei. Of note, some cells yet displayed intermediate phenotypes (Fig 3E, see close-ups), reinforcing the conclusion that Svb^{ACT} actively counteracts differentiation. Finally, *singed* that encodes Fascin, an actin-bundling protein strongly upregulated in epithelial tumors, is a direct target of Svb in epidermal cells (Chanut-Delalande *et al*, 2006) and the *snE1* enhancer provides readout of Svb^{ACT} activity (Menoret *et al*, 2013). We found that *snE1* was specifically expressed in ISC/EBs and mutations of Svb-binding sites abrogated *snE1* activity in the midgut (Fig EV3D). These results thus provide conclusive evidence that Svb behaves as an activator in intestinal stem cells, further suggesting that it regulates the expression of cytoskeleton and cell junction factors, as in

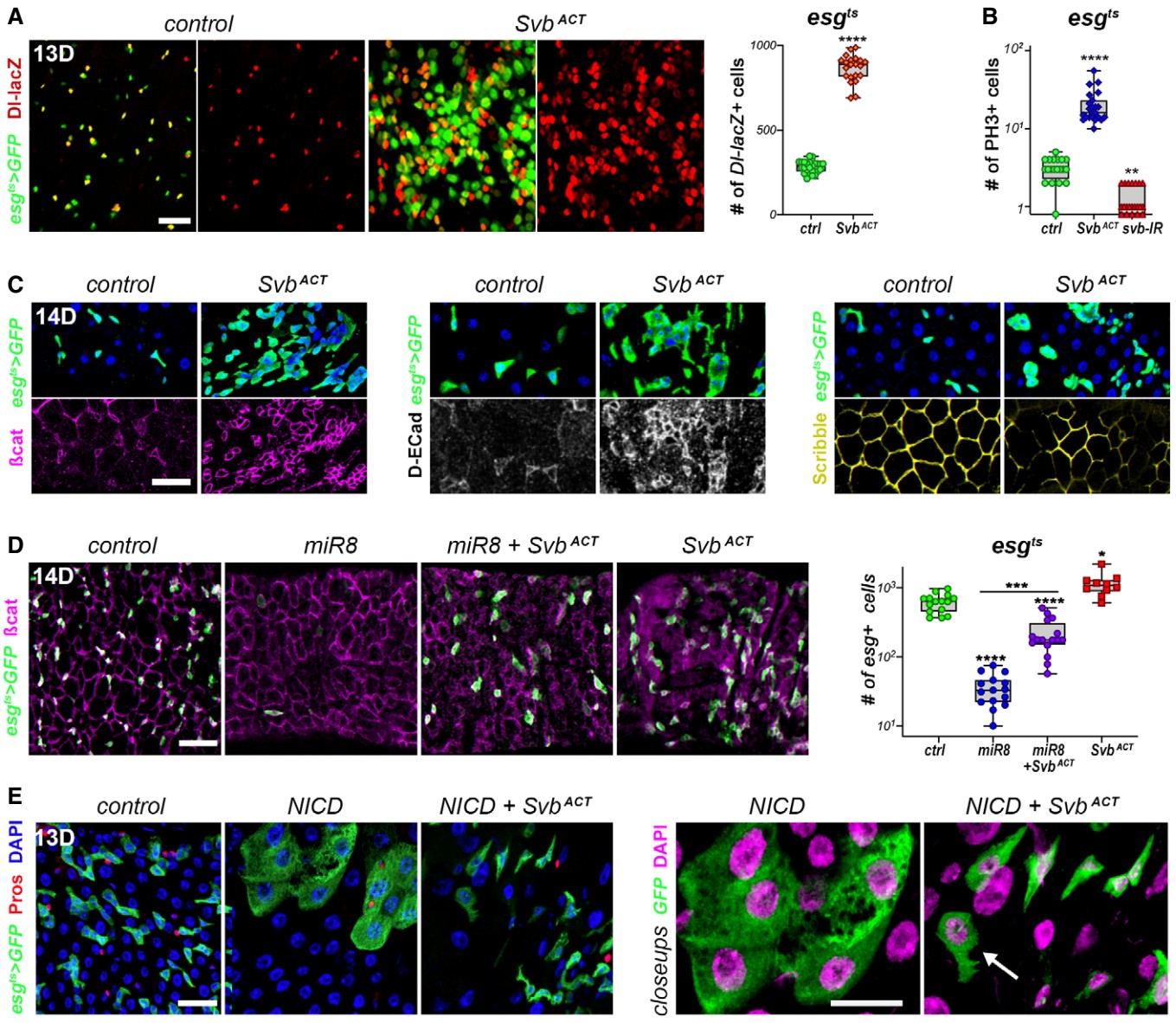

**Figure 3. Svb activator induces stem cell proliferation.**

A   *esg^ts* midguts expressing GFP alone (control), or expressing OvoB, and carrying a *Dl-lacZ* transgene that marks ISCs. Samples were stained for GFP (green) and β-Gal (red); the graph shows quantification of β-Gal-positive cells (ISCs).

B   Quantification of number of mitotic PH3-positive cells/midgut in *esg^ts* guts expressing GFP alone (control), or expressing OvoB and *svb*-RNAi; y-axis is drawn as log(10).

C   *esg^ts* midguts expressing GFP alone (control) or expressing Svb^ACT. Samples were stained for GFP (green), and β-catenin (purple), DE-Cadherin (white) or Scribble (yellow). Top and bottom pictures show separate channels for a same region.

D   *esg^ts* midguts expressing GFP alone (control), or expressing *miR8*, OvoB, and *mir8*+ OvoB. Samples were stained for GFP (green) and β-catenin (purple). The graph shows quantification of GFP-positive cells for each genotype. The y-axis is drawn using a log(10) scale.

E   *esg^ts* midguts expressing GFP alone (control), or expressing Notch Intra Cellular Domain (NICD), and NICD+ OvoB. In close-ups (right), DAPI is shown in purple for improved contrast; the arrow highlights a cell with intermediate phenotype.

Data information: Boxes extend from the 25th to 75th percentiles, whiskers from min to max, the line in each box is plotted at the median; data were collected from three independent replicates. *P* values from Mann–Whitney tests (A) and one-way ANOVA (B,D) are: * < 0.05, ** < 0.01, *** < 0.001, **** < 0.0001. Blue is DAPI. Scale bars are 20 μm.

embryonic epithelial cells (Chanut-Delalande *et al*, 2006; Fernandes *et al*, 2010; Menoret *et al*, 2013).

Taken together, our data show that Svb^ACT is sufficient to induce characteristics of stem cells such as typical cellular architecture and proliferative capability, and to prevent differentiation.

**Svb mediates Wnt and EGFR mitogenic pathways in intestinal stem cells**

During embryogenesis, Svb mediates the activity of EGFR and Wnt signaling pathways for epidermal differentiation (Payre *et al*, 1999;

Payre, 2004). Since these pathways are key regulators of normal and cancer stem cells (Li & Jasper, 2016; Perochon *et al*, 2018), we investigated their putative relationship with Svb in intestinal stem cells.

EGFR and Wnt are the main mitogenic pathways in the intestine under homeostatic conditions. Upregulation of EGFR ($esg^{ts}F/O > Ras^{V12}$) or Wnt ($esg^{ts} > Arm^{S10}$) leads to ISC proliferation (Lin *et al*, 2008) and *svb* knockdown was sufficient to suppress these phenotypes, resulting in a strongly decreased population of ISC/EBs (Fig 4A and B). Therefore, the mitogenic activity of Wnt and EGFR in stem cells requires *svb* function. Furthermore, we found that Svb$^{ACT}$ was capable to induce ISC hyperproliferation even when these pathways were blocked. Inhibition of EGFR ($esg^{ts} > $ EGFR-DN) or Wnt ($esg^{ts} > $ TCF-DN) induces a marked loss of ISC/EBs (Lin *et al*, 2008) and, in both cases, the expression of Svb$^{ACT}$ rescued these phenotypes, still leading to a twofold-threefold increase in stem/progenitor population when compared to controls (Fig 4A and B). Hence, Svb is epistatic to, in other words is a downstream effector of, Wnt and EGFR pathways and mediates their activity for stem cell maintenance and self-renewal.

To get comprehensive insight into the mechanisms linking mitogenic pathways to the control of *svb* expression, we undertook an *in vivo* screen to identify transcription factors that regulate *svb* enhancers (see Materials and Methods). Following the individual inactivation of 220 candidates, the two top list factors were Pointed (Pnt), an *ets* effector of the EGFR pathway, and TCF, the nuclear effector of Wnt. The *E3N* sequence contains putative binding sites for both Pnt and TCF (Figs 4C and EV4A), suggesting that they directly activate *E3N* expression in ISC/EBs. To confirm this, we generated *E3N* variants-bearing mutations that inactivate Pnt (*E3N-Pnt-mt*) or TCF (*E3N-TCF-mt*)-binding sites. Both *E3N-Pnt-mt* and *E3N-TCF-mt* displayed strongly decreased activity (Fig EV4B and C), leading to barely detectable expression in ISC/EBs (Fig 4C). Therefore, the binding of Pnt and TCF appears critical for the function of the *E3N* enhancer that drives *svb* transcription in ISC/EBs.

These results support the conclusion that Svb is a direct downstream target of Wnt and EGFR in adult ISC/EBs and integrates local signaling pathways to endorse renewal and stemness of intestinal progenitors.

## Svb repressor promotes differentiation into enterocytes

In addition to Svb$^{ACT}$ in ISC/EBs, we next explored the putative role of Svb$^{REP}$ and whether Svb was also active in later stages of the intestinal lineage.

*In situ* hybridization confirmed *svb* expression in the adult intestine. Basal views of the intestinal epithelium showed *svb* mRNA accumulating in ISC/EBs, which are seen as small doublet cells apposed to the basement membrane (Fig 5A). Apical views further revealed *svb* expression in ECs, characterized by their very large polyploid nuclei (Fig 5A). Svb expression in both ISC/EBS and ECs was also confirmed by analysis of a *svb::GFP* mini-gene rescue construct (Menoret *et al*, 2013). The Svb::GFP protein was detected in ISC/EBs and in ECs (Fig 5B–B′), but not in EEs (that are not affected by *svb* loss of function, see Fig EV1). As aforementioned, *svb* expression in ECs is driven by a separate enhancer, called *7* (Fig 1A). Within *svb* enhancer *7*, we delineated a minimal region, *9CJ2* (232 bp), that drives specific expression in ECs (Fig 5C). Therefore, *svb* expression

in the intestinal lineage relies on two distinct enhancers: *E3N* in stem/progenitor cells and *9CJ2* in enterocytes.

The differential expression of *svb* enhancers implied that they capture different regulatory inputs. We used our *in vivo* screen to identify factors responsible for *9CJ2* activity and found that Pdm1 (a.k.a. Nubbin) is critical for *9CJ2* function. Interestingly, Pdm1 is a conserved POU factor that is a hallmark of ECs (Jiang *et al*, 2009; Beebe *et al*, 2010). There are two putative Pdm1-binding sites within the *9CJ2 svb* enhancer (Fig EV4D), which we inactivated by point mutations (*9CJ2-Pdm-mt*). Knockout of Pdm1 sites disrupted *9CJ2* activity (Figs 5C and EV4E), supporting that *svb* expression in ECs is under direct control of the enterocyte factor Pdm1.

The switch in Svb transcriptional activity triggered by Pri peptides is associated with a marked change in Svb intranuclear distribution: Whereas Svb$^{ACT}$ diffuses within the nucleoplasm, Svb$^{REP}$ accumulates in dense foci (Kondo *et al*, 2010; Zanet *et al*, 2015). We found that Svb is diffused in $esg^+$ cells that express *pri*, while displaying foci in ECs, which do not (Fig 5B′). Hence, unlike ISCs that rely on Svb$^{ACT}$, later stages of the intestinal lineage were likely to involve Svb$^{REP}$. To test this, we assayed consequences of expressing Svb$^{REP}$ using the $esg^{ts}$ system. The number of ISC/EBs was markedly reduced and remaining $esg^+$ cells displayed aberrant morphology (Fig 5D). These $esg^+$ cells were larger and their nuclei were significantly bigger than nuclei of wild-type ISCs (see close-ups Fig 5D). Svb$^{REP}$ also severely reduced the growth of $esg^{ts}F/O$ clones, which contained individual cells with large nuclei (Fig 5E). These results led us to hypothesize that Svb$^{REP}$ causes stem cell loss through precocious differentiation rather than cell death. Indeed, GFP$^+$ cells of $esg^{ts}F/O > Svb^{REP}$ intestines were negative for Dcp1 apoptotic staining (Fig 5E) and DIAP1 overexpression did not suppress Svb$^{REP}$ phenotypes (Fig 5E′). These data ruled out stem cell apoptosis and lineage tracing fully supported the notion that Svb$^{REP}$ induces massive differentiation. When Svb$^{REP}$ was expressed in $esg^+$ cells using the ReDDM system, the loss of ISC/EBs was accompanied by a strong increase in their differentiated progeny (Fig 5F). We also observed that enlarged Svb$^{REP}$ cells that still express low levels of *esg*-GFP became positive for Pdm1 (Fig 5G), indicating that they engaged precocious differentiation. Thus, Svb$^{REP}$ is sufficient to trigger a loss of stem cell identity and results in the initiation of EC differentiation.

A main determinant of ISC differentiation is the activation of Notch. We thus assayed whether the differentiation potential of Svb$^{REP}$ relied on Notch and/or other regulatory pathways of intestinal stem cells. Inhibition of Notch ($esg^{ts} > $ Notch-RNAi) induces dramatic tumor-like expansion of ISCs (Ohlstein & Spradling, 2007). Strikingly, co-expression of Svb$^{REP}$ was sufficient to suppress Notch-deficient tumors and enforce differentiation, as manifested by enlarged GFP$^+$ cells with big nuclei (Fig 6A). Svb$^{REP}$ also suppressed ISC-derived tumors resulting from the inactivation of JAK/STAT (Fig 6B), which also regulates differentiation of the intestinal lineage (Buchon *et al*, 2009; Jiang *et al*, 2009). Finally, Svb$^{REP}$ was able to suppress stem cell hyperplasia triggered by Wnt overactivation (Fig 6C). These results well illustrate that Svb$^{REP}$ forces tumor cells to differentiate, as seen by prominent changes in morphology and increased nuclear size. Of note, these phenotypes were strikingly different from those observed for *svb* loss of function (Fig 6C), which prevents stem cell overgrowth but cannot impose differentiation. Hence, Svb$^{REP}$ acts as a potent tumor suppressor, sufficient to

impose differentiation and prevent stem cell proliferation triggered by altered signaling.

In sum, *svb* enhancers directly integrate different regulatory inputs to drive specific expression either in stem cells or in enterocytes. While Svb[ACT] promotes stemness and proliferation, these results demonstrate that Svb[REP] drives enterocyte differentiation, in both normal and tumorous contexts.

## Svb[REP] is required to maintain enterocyte differentiation, and Svb[ACT] triggers hallmarks of dedifferentiation

We further investigated the role of Svb isoforms in differentiated ECs. To avoid indirect consequences linked to expression in stem/progenitor cells, we used the temperature-sensitive driver *MyoIA*[ts] (Jiang *et al*, 2009). *MyoIA* encodes a gut-specific myosin that is a

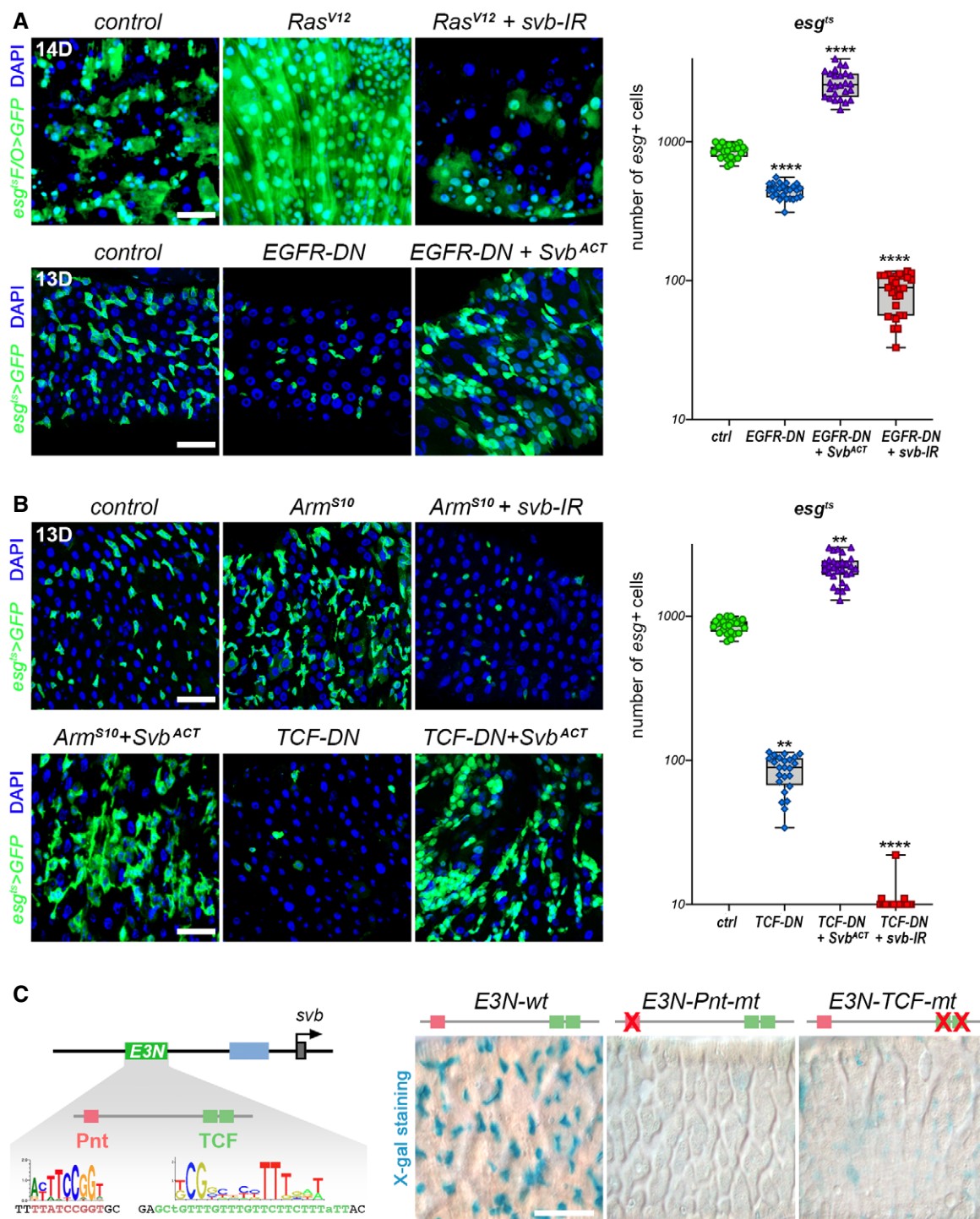

**Figure 4.**

◄ **Figure 4. Svb acts downstream of Wnt and EGFR mitogenic signaling pathways in the adult midgut.**

A   $esg^{ts}F/O$ midguts expressing GFP alone (control), or expressing RasV$^{12}$, and RasV$^{12}$+ $sub$-RNAi (top panels). Bottom panels show $esg^{ts}$ midguts expressing GFP alone (control), or expressing EGFR-DN, and EGFR-DN+ OvoB. Samples were stained for GFP (green). The graph shows quantification of the number of GFP-positive cells in $esg^{ts}$ midguts expressing GFP alone (ctrl), or expressing EGFR-DN, EGFR-DN+ OvoB, and EGFR-DN+ $sub$-RNAi. The y-axis is plotted as log(10).

B   $esg^{ts}$ midguts expressing GFP alone (control), or expressing Arm$^{S10}$, Arm$^{S10}$+ $sub$-RNAi, Arm$^{S10}$+ OvoB, TCF-DN, and TCF-DN+ OvoB. Samples were stained for GFP (green). The graph shows quantification of the number of GFP-positive cells in $esg^{ts}$ midguts expressing GFP alone (ctrl), or expressing TCF-DN, TCF-DN+ OvoB, and TCF-DN+ $sub$-RNAi. The y-axis is plotted as log(10).

C   The drawing at left schematizes the $sub$ locus, with position of the E3N enhancer. Close-up shows E3N sequence that correspond to binding sites for Pnt (red) and TCF (green). Right subpanels are pictures of posterior midguts showing expression of wild-type $E3N-lacZ$ ($E3N\omega t$), or $E3N-Pnt-mt$, and $E3N-TCF-mt$, as seen from X-Gal staining (cyan blue).

Data information: Boxes extend from the 25$^{th}$ to 75$^{th}$ percentiles, whiskers from min to max, the line in each box is plotted at the median; data were collected from three independent replicates. $P$ values from one-way ANOVA are ** < 0.01, **** < 0.0001. In all pictures of (A and B) panels, blue is DAPI. Scale bars are 20 μm.

component of the apical brush border and found only in differentiated enterocytes.

Knocking down $svb$ in ECs ($MyoIA^{ts} > svb$-RNAi) led to a gross alteration of the midgut, with a thinner epithelium and enlarged lumen (Fig 7A). The lack of $svb$ also impaired EC differentiation, as $MyoIA$-GFP expression was decreased (Fig 7B). Elevated Dcp1 levels were suggestive of increased apoptosis (Fig EV5A), as also supported by ultra-structural analyses showing pyknotic nuclei and defective cell contacts (Fig 7C). As observed for ISC/EBs, $svb$ also prevents apoptosis of mature ECs, in which $svb$ function is further required to maintain differentiation.

The pattern of Svb::GFP intranuclear distribution suggested that Svb was acting as a repressor in ECs (see Fig 5). We tested this hypothesis through a series of complementary experiments. While $MyoIA^{ts} > Svb^{REP}$ intestines showed no detectable homeostatic or structural changes (Fig EV5A–C), forced expression of Svb$^{ACT}$ in ECs had dramatic effects on the midgut, with abnormal multilayered intestinal epithelium and a reduced lumen (Fig 7A and D). It also caused loss of $MyoIA$-GFP, indicating deeply compromised differentiation (Fig 7B), as also manifested by disruption of brush border microvilli (Fig 7C). The dramatic phenotypes observed upon 2 weeks of induction prompted us to use shorter treatments (6 days). Even in these milder conditions, Svb$^{ACT}$ disrupted the intestinal epithelium, with multilayered cells displaying reduced apical actin and altered organization, as highlighted by staining for Tsp2a or Coracle (Fig 7D). The conclusion that $svb$ function in ECs relies on the unprocessed Svb$^{REP}$ raised specific predictions, which we assayed directly. First, unlike in stem cells, Svb activity in ECs should not depend on factors that operate its processing into the activator, i.e., it should be insensitive to the loss of $Ubr3$. Accordingly, we did not detect defects upon $Ubr3$ knockdown in ECs, intestines exhibiting proper levels of GFP and organization (Fig EV5B). Second, since $pri$ is normally absent from ECs (Fig 2), forced expression of $pri$ should trigger processing of endogenous Svb$^{REP}$ into Svb$^{ACT}$. Indeed, $MyoIA^{ts}$-driven expression of $pri$ in ECs induced defects resembling those seen with Svb$^{ACT}$, albeit of weaker severity (Fig 7D). Hence, these data demonstrate that the repressor form of Svb is required to maintain differentiation of ECs.

We then investigated in more details the phenotypes caused by Svb$^{ACT}$ in enterocytes, which were stronger than the loss of $svb$. As seen in stem cells (Fig 3), expression of Svb$^{ACT}$ in ECs led to remodeling of the epithelial architecture, featured by basolateral accumulation of β-Catenin, and decreased Scribble in lateral domains (Fig 8A). Close inspection revealed that large polyploid EC-like cells

with reduced or undetectable GFP levels remained in the gut following induction of Svb$^{ACT}$ in ECs. Some cells displayed extreme phenotypes, with massive accumulation of β-catenin and withdrawal of Scribble (Fig 8A′). Svb$^{ACT}$ also induced over-proliferation, with high increase in the number of PH3$^{+}$ intestinal cells (Fig 8B). These mitotic cells were likely ISCs, since damaged or dying ECs produce short-range signals, such as Upd1-3 cytokines, which foster regenerative proliferation of neighbor stem cells (Buchon et al, 2009; Jiang et al, 2009). However, we observed some PH3$^{+}$ cells that also express $Myo1A^{ts}$-GFP (Fig 8C), suggesting that Svb$^{ACT}$ can force late EBs or ECs to reenter the cell cycle.

Therefore, our data show the importance of Svb$^{REP}$ to trigger and maintain enterocyte differentiation. Furthermore, the proper regulation of Svb processing is crucial, since ectopic production of Svb$^{ACT}$ in ECs induces loss of differentiation markers and gain of features normally seen in stem cells.

# Discussion

Our data show that the OvoL/Shavenbaby transcription factor is a key integrator of intrinsic, local, and systemic cues to control the behavior of adult intestinal stem cells and of their progeny. In stem cells, Svb is processed into the activator form (Svb$^{ACT}$) that mediates EGFR and Wnt activities for stem cell self-renewal. Pdm1 then drives $svb$ expression in later stages of the lineage, during which Svb behaves as a repressor (Svb$^{REP}$) that direct differentiation into enterocytes. The balance between Svb$^{ACT}$ and Svb$^{REP}$ is gated by Pri peptides, which allow conversion of Svb transcriptional activity in response to systemic ecdysone signaling. These results show the pivotal role of Svb in balancing stem cell renewal/proliferation versus differentiation, and further suggest that OvoL factors are evolutionarily conserved determinants of stemness.

## $sub$ integrates multiple regulatory cues for the homeostasis of adult stem cells

Numerous studies have demonstrated the role of Wnt and EGFR signaling pathways in somatic stem cells and cancers (Normanno et al, 2006; Zhan et al, 2017). In the $Drosophila$ intestine, EGFR pathway acts in an autocrine/paracrine manner to promote homeostatic stem cell self-renewal (Jiang & Edgar, 2009; Biteau & Jasper, 2011; Li & Jasper, 2016), whereas Wnt signals are mainly produced by visceral muscles that act as a niche (Perochon et al, 2018) (see

Fig 8D). We find that *ovo/svb* is a common target of Wnt and EGFR in adult stem cells that mediates their activity to promote stem cell self-renewal. Our data further indicate that the nuclear mediators of Wnt (TCF) and EGFR (Pointed) activate *svb* expression in stem cells, through direct regulation of an enhancer (*E3N*) driving ISC/EB-specific expression. Although the precise register of Wnt activity in

the midgut remains to be confirmed (Perochon *et al*, 2018), the main mitogenic pathway EGFR is high in ISC/EBs and strongly reduced in ECs (Jiang & Edgar, 2009; Jin *et al*, 2015), explaining specific expression of *E3N* in stem and progenitor cells (Fig 8D).

A separate regulatory module consisting of the *9CJ2 svb* enhancer, under direct control of the POU transcription factor Pdm1

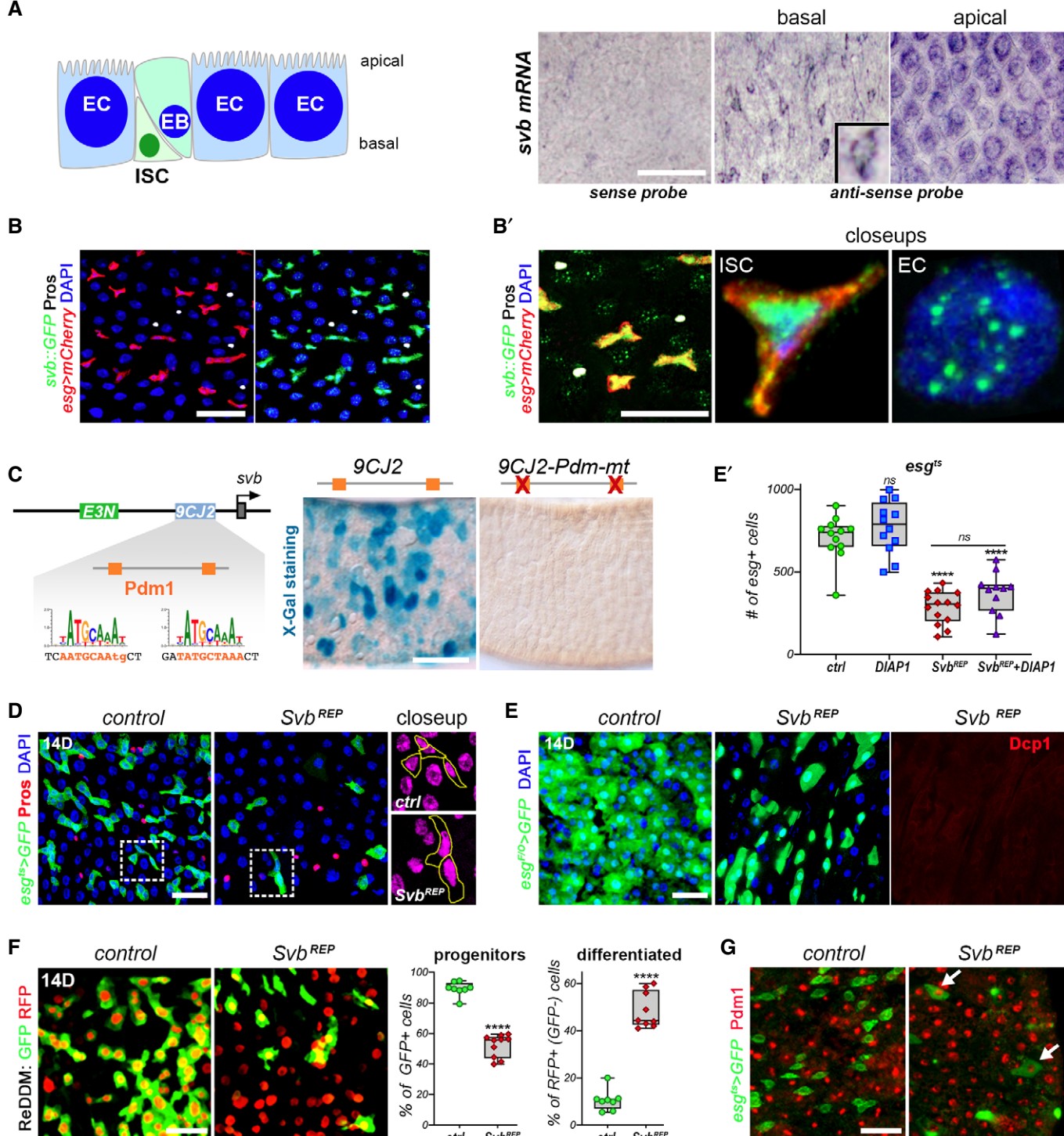

**Figure 5.**

**Figure 5.  Svb repressor promotes enterocyte differentiation.**

A   Drawing of apical–basal organization of the intestinal epithelium and expression of *svb* mRNA as revealed by *in situ* hybridization. The inlet shows an enlarged view.
B   *esg-Gal4* midguts expressing mCherry and a *svb::GFP* rescue mini-gene, consisting of *svb* cDNA tagged by GFP (green) and driven by E and 7 *svb* enhancers (see Fig 1). Samples were stained for GFP (green), mCherry (red), and Prospero (white). (B′) shows close-up views.
C   The drawing at left schematizes the *svb* locus, with position of *E3N* and *9CJ2* enhancers. Close-up shows *9CJ2* sequence with binding sites for Pdm1 (orange). The right subpanels are posterior midguts showing expression of wild-type *9CJ2 svb* enhancer, and expression of *9CJ2-Pdm-mt* in which Pdm1-binding sites have been mutated.
D   *esg*ts midguts expressing GFP alone (control), or expressing Svb REP. Samples were stained for GFP (green) and Prospero (red). Close-ups correspond to boxed regions, with DAPI shown in purple and GFP-positive cells outlined in yellow.
E   *esg*ts F/O midguts expressing GFP alone (control), or expressing Svb REP. Samples were stained for GFP (green) and cleaved DCP1 (red). (E′) shows quantification of the number of GFP-positive cells in *esg*ts guts expressing GFP alone (ctrl), or DIAP, Svb REP, and Svb REP + DIAP.
F   ReDDM lineage tracing in control midguts, or in midguts expressing Svb REP, and quantification of the percentage of progenitors (GFP-positive, RFP-positive) *versus* differentiated cells (GFP-negative, RFP-positive).
G   *esg*ts midguts expressing GFP alone (control), or expressing Svb REP. Samples were stained for GFP (green) and Pdm1 (red); arrows show enlarged GFP-positive cells which are also positive for Pdm1.

Data information: Boxes extend from the 25th to 75th percentiles, whiskers from min to max, the line in each box is plotted at the median; data were collected from three independent replicates. *P* values from one-way ANOVA (E′) and Mann–Whitney tests (F) are: ns > 0.5, **** < 0.0001. Blue is DAPI, scale bars, 20 μm.

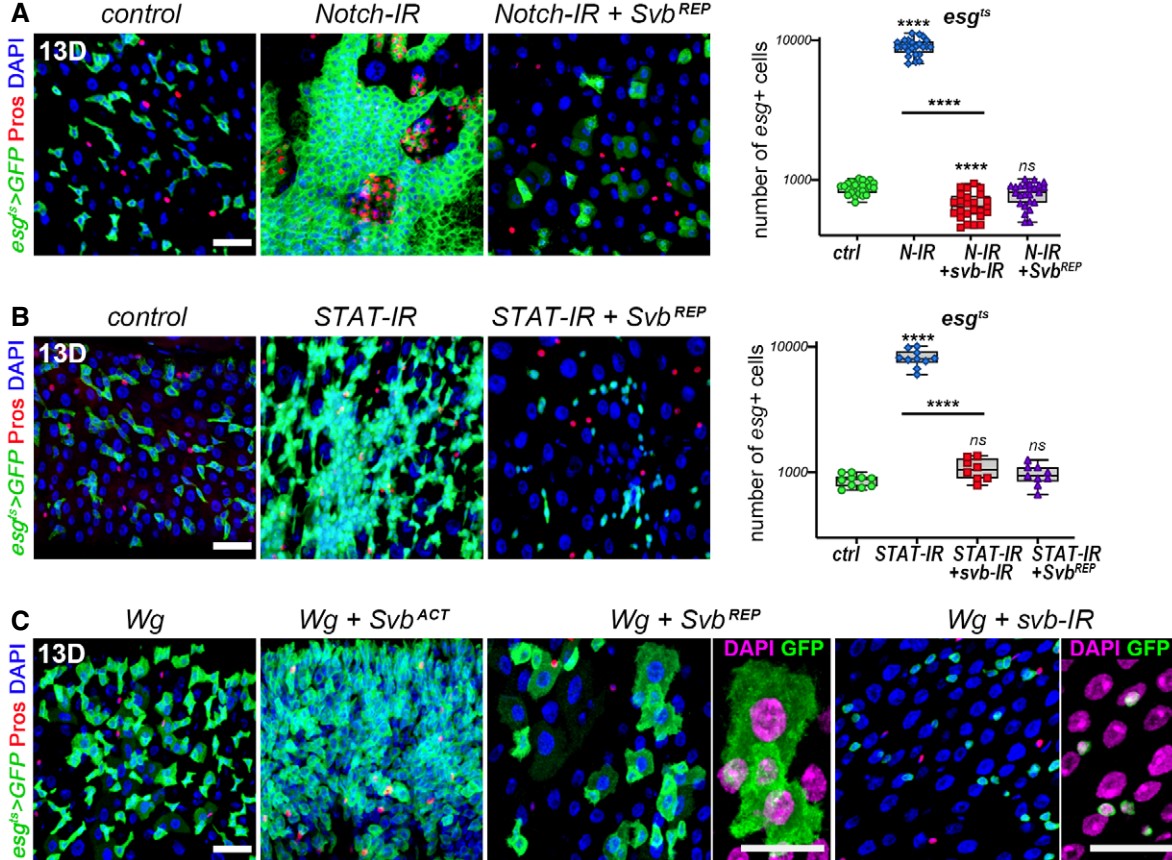

**Figure 6.  Svb REP suppresses stem cell tumors in the gut epithelium.**

A   *esg*ts midguts expressing GFP alone (control), or expressing *Notch*-RNAi, *Notch*-RNAi+ *svb*-RNAi, and *Notch*-RNAi+ Svb REP. Samples were stained for GFP (green) and Prospero (red). The graph shows quantification of the number of GFP-positive cells.
B   *esg*ts midguts expressing GFP alone (control), or expressing *STAT*-RNAi, *STAT*-RNAi+ *svb*-RNAi and *STAT*-RNAi+ Svb REP. Samples were stained for GFP (green) and Prospero (red). The graph shows quantification of the number of GFP-positive cells.
C   *esg*ts midguts expressing GFP alone (control), or expressing Wg, Wg+ OvoB, Wg+ Svb REP, and Wg+ *svb*-RNAi. In close-up views, nuclei are in purple.

Data information: Boxes extend from the 25th to 75th percentiles, whiskers from min to max, the line in each box is plotted at the median, data were collected from three independent replicates. *P* values from one-way ANOVA are: ns *P* > 0.5, **** < 0.0001. Graphs are drawn using a log(10) y-axis scale. DAPI is blue; scale bars, 20 μm.

(Fig 8D), maintains *svb* expression in later stages of the lineage. Pdm1 is highly expressed in differentiated enterocytes (Jiang *et al*, 2009; Beebe *et al*, 2010) and is a main marker of mature ECs (Li &

Jasper, 2016). Since there is evidence for mutual antagonism between Pdm1 and Escargot (Korzelius *et al*, 2014; Tang *et al*, 2018), Escargot might repress *Pdm1* expression in stem/progenitor

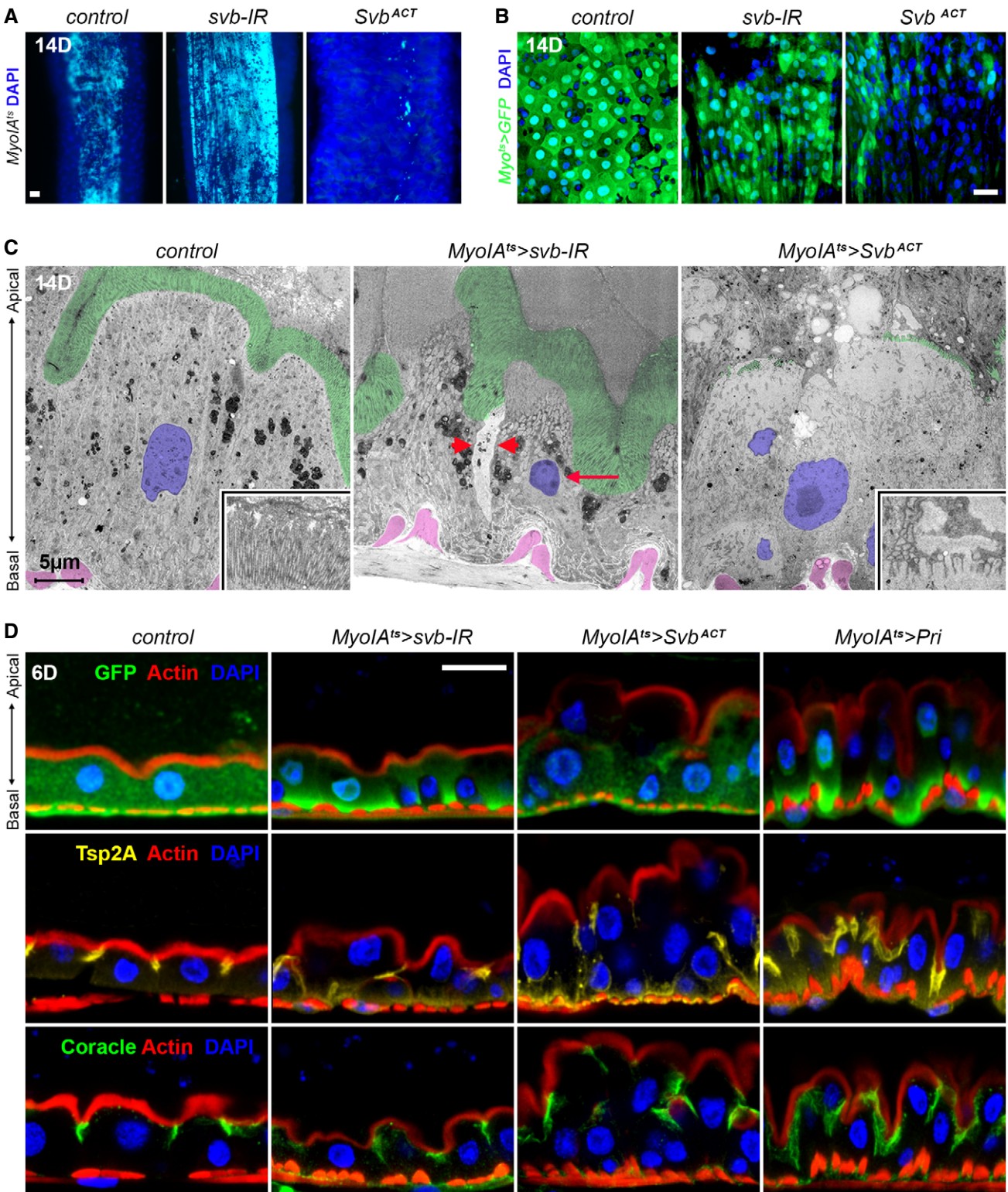

**Figure 7.**

◀

**Figure 7.  Svb repressor is required to maintain enterocyte differentiation.**

A   Control *MyoIA^{ts}* midguts, and *MyoIA^{ts}* > *sub*-RNAi or *MyoIA^{ts}* > OvoB midguts. Cyan dye stains the lumen.

B   *MyoIA^{ts}* midguts expressing GFP alone (control), or expressing *sub*-RNAi, and Svb^{ACT}. Samples were stained for GFP (green).

C   Electron micrographs of *MyoIA^{ts}* control midguts, or expressing *sub*-RNAi and Svb^{ACT}. Brush border microvilli are pseudo-colored in green, and high magnification views are shown in inlets. Nuclei are pseudo-colored in purple, and visceral muscles located above the basement membrane are in pink. Arrowheads point to impaired cell contacts, the arrow points to a pyknotic nucleus.

D   Cross sections of control *MyoIA^{ts}* midguts (expressing GFP and *mCherry*-RNAi), or expressing *sub*-RNAi, Svb^{ACT}, and Pri. Samples were stained for F-actin (white), GFP (green) and Tsp2a (yellow).

Data information: Blue is DAPI. Scale bars are 20 μm (A, B, D) and 5 μm in (C).

cells and, thereby, would restrict activity of the *9CJ2 svb* enhancer to enterocytes.

*E3N* and *9CJ2* enhancers also drive *svb* expression in the embryo, supporting the notion that they harbor pleiotropic functions across the life-cycle (Preger-Ben Noon *et al*, 2018). Both *svb* enhancers capture additional regulatory cues during development, including Hox proteins and Exd/Hth cofactors (Crocker *et al*, 2015), as well as GATA and LIM homeodomain factors in the case of *E3N* (Preger-Ben Noon *et al*, 2016), opening the possibility that these factors could as well regulate intestinal stem cells in the adult. *svb* expression in the adult midgut also involves the *E6 svb* enhancer, which like *E3N* is active in ISC/EBs. Apparently redundant *svb* enhancers may ensure robustness of intestinal homeostasis in the face of genetic or environmental variations, as shown for epidermal development (Frankel *et al*, 2010). Together, our data provide mechanistic information on how the *svb* cis-regulatory landscape integrates multiple cues to drive stage-specific expression in the intestinal stem cell lineage (Fig 8D).

## Svb^{ACT}: a key factor for stemness and stem cell renewal

In addition to the control of *svb* expression, activity of the Svb factor is tightly regulated by post-translational modification, which relies on proteasome-mediated processing (Kondo *et al*, 2010; Zanet *et al*, 2015). We show that Svb processing into an activator is indispensable to maintain and prevent differentiation of intestinal stem cells, which otherwise undergo apoptosis, as recently reported for renal nephric stem cells (Bohere *et al*, 2018). Although renal stem cells are mostly quiescent (Bohere *et al*, 2018; Xu *et al*, 2018), intestinal stem cells self-renew under homeostatic conditions and proliferate in response to various challenges (Li & Jasper, 2016). This high plasticity of the intestinal lineage further reveals that Svb^{ACT} is both required and sufficient to promote stem cell proliferation; high Svb^{ACT} in ISCs leading to hyperplasic overgrowth (Fig 8D). Supernumerary cells induced by Svb^{ACT} display typical features of stem cells, including redistribution of cell junction and apical-basal polarity complexes. These data support a model in which Svb^{ACT} might be an intrinsic determinant of stemness. Furthermore, forced expression of Svb^{ACT} strongly alters ECs, which lose differentiation and in some cases engage mitosis. Similar epithelial dysplasia progressively appears when the gut experiences aging (Biteau *et al*, 2008; Biteau *et al*, 2010). Future studies will determine whether Svb mis-regulation is involved in aging, and/or if Svb^{ACT} is capable to induce dedifferentiation.

## Svb^{REP} triggers enterocyte differentiation

Our results show the function of Shavenbaby in enterocytes, but not in enteroendocrine cells, consistent with an early separation between EC and EE lineages (Biteau & Jasper, 2014; Guo & Ohlstein, 2015; Zeng & Hou, 2015). In contrast to stem cells, Svb acts as a repressor within ECs in which it is required for their maintenance and differentiation. Ectopic processing of Svb in ECs disrupts epithelial organization, leading to multilayered cells that lose features of mature ECs, including brush border microvilli, as well as properly organized cell–cell junctions. Junctional complexes are progressively established during EB to EC maturation and they are essential for differentiation. For instance, the septate junction component Tsp2A is required for downregulation in ECs of Hippo and JAK/STAT signaling, which otherwise promote proliferation (Xu *et al*, 2019). Likewise, Svb^{REP} promotes EC differentiation and is also a potent inhibitor of stem cell proliferation. This is the case under homeostatic conditions and, importantly, Svb^{REP} can also suppress hyperproliferation of stem cells induced by altered Notch, STAT, Wnt, or EGFR signaling (Fig 8D). Of note, Svb^{REP} enforced tumor cell differentiation, while *svb* loss of function prevents stem cell growth but does not induce differentiation. Therefore, Svb^{ACT} and Svb^{REP} exert antagonistic functions within the adult intestinal lineage, Svb^{ACT} promoting stem cell survival and proliferation, while Svb^{REP} later acts to induce and maintain enterocyte differentiation.

## Ecdysone function in intestinal stem cells

Throughout development, the maturing processing of Svb is triggered by Pri peptides (Kondo *et al*, 2010; Chanut-Delalande *et al*, 2014; Zanet *et al*, 2015; Bohere *et al*, 2018). In the adult intestine, *pri* is specifically expressed in ISC/EBs, and Pri peptides are required—with their target Ubr3 ubiquitin ligase—for stem cell maintenance. Previous findings have led us to propose that a key role of Pri is to mediate ecdysone signaling to implement systemic hormonal control within gene regulatory networks, as seen for developmental timing of epidermal derivatives (Chanut-Delalande *et al*, 2014). Consistent with this view, we show that inactivation of the ecdysone receptor EcR within intestinal stem cells and enteroblasts strongly impacts their behavior, decreasing proliferation and promoting differentiation, *i.e.*, as seen upon inhibition of Svb processing. These results were particularly surprising because ecdysone is not produced in the gut, ovaries being the major source of ecdysone in adult females after mating (Uryu *et al*, 2015; Ahmed *et al*, 2020). Thus, they imply the existence of sex-specific inter-organ communication that regulates the fate of somatic stem cells, a feature that has never been reported so far, to our best knowledge. Two contemporary studies confirm the role of ecdysone in sustaining stemness and undifferentiated state of *Drosophila* ISCs in the midgut. Both studies demonstrate that EcR and its cofactor Usp foster division and expansion of ISCs in response to a peak of

steroids synthesized in ovaries upon mating (Ahmed *et al*, 2020; Zipper *et al*, 2020). These data provide compelling evidence for ovary-to-gut communication and show that sex hormones remodel

stem cell fate to adjust organ size, as means to face elevated energetic costs imposed by reproduction. Because the expression of *pri*, or of Svb^ACT, can overcome EcR inactivation in ISC/EBs, our data

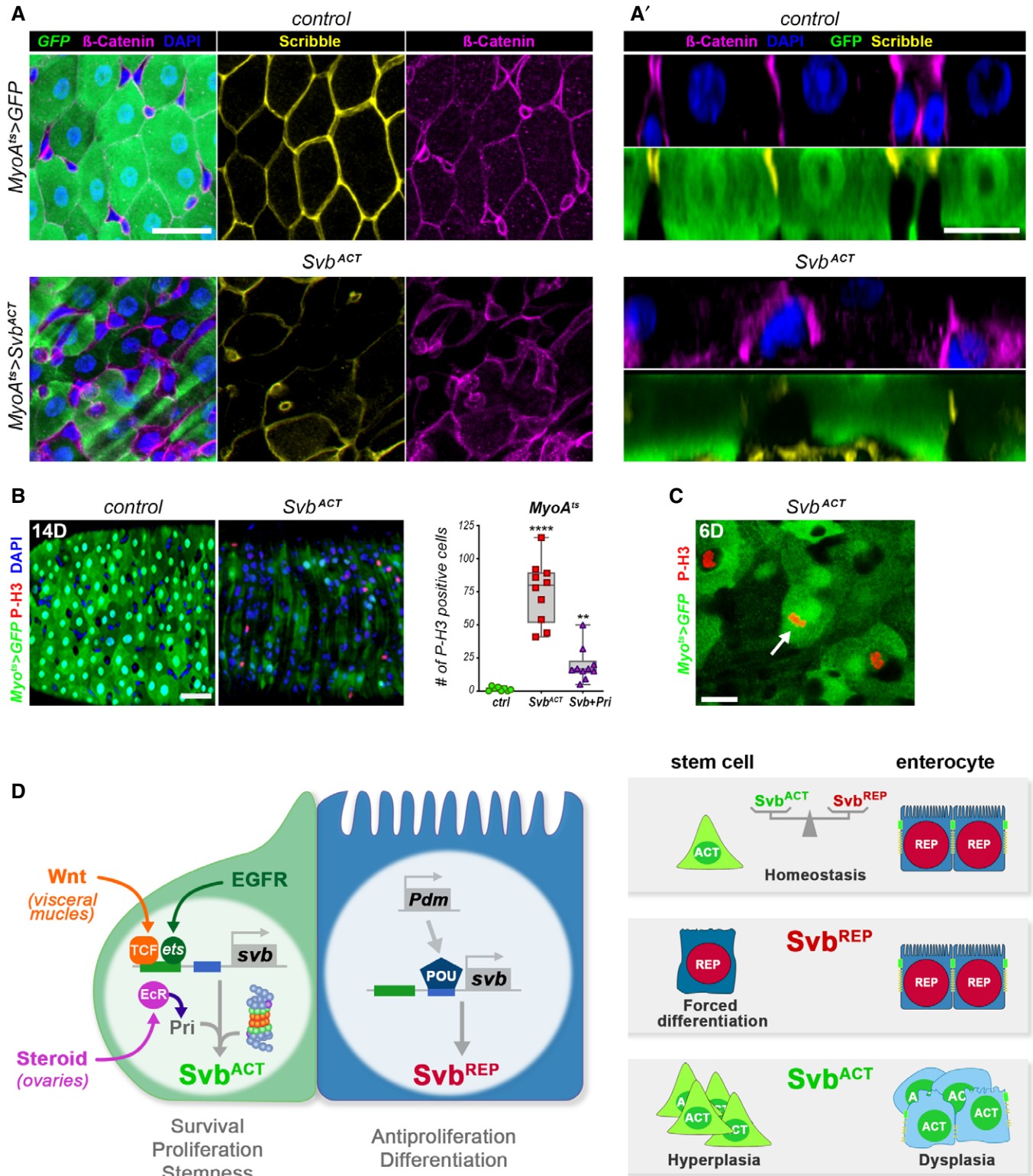

**Figure 8.**

**Figure 8. Ectopic Svb processing disrupts enterocyte differentiation.**

A   *MyoIA*[ts] midguts expressing GFP alone (control), or expressing Svb[ACT]. Samples were stained for GFP (green), Scribble (yellow), β-catenin (purple), and DAPI (Blue). (A') pictures display cross sections of the regions shown in (A).

B   *MyoIA*[ts] midguts expressing GFP alone (control), or expressing Svb[ACT]. Samples were stained for GFP (green) and PH3 (red). The graph plots number of mitotic PH3-positive cells *per* midgut of *MyoIA*[ts] guts expressing GFP alone (ctrl), or expressing Svb[ACT], and Svb[REP]+ *pri*.

C   *MyoIA*[ts] midguts expressing GFP and Svb[ACT] (green), and stained for GFP (green) and PH3 (red). The picture is a single focal plane; the arrow shows a large GFP-positive cell, which is also positive for mitotic PH3.

D   Summary of the role of Svb[ACT] and Svb[REP] in the control of intestinal stem cell maintenance, proliferation, and differentiation.

Data information: Boxes extend from the 25th to 75th percentiles, whiskers from min to max, the horizontal line in each box is plotted at the median; data were collected from three independent replicates. *P* values from one-way ANOVA are: ** <0.01, **** <0.0001. Scale bars are 20 μm.

suggest that the activation of *pri* to increase Svb[ACT] levels is a nexus target of steroid action in intestinal stem cells.

### OvoL/Svb transcriptional switch for stem cell control across animals

Mounting evidence suggests a wide role of OvoL/Svb factors in progenitor and stem cells across animals. Unlike *Drosophila*, most insects develop by sequential addition of posterior segments, from a group of embryonic precursors referred to as posterior growth zone. In such species, Svb is specifically expressed in these precursors and required for the formation of posterior structures, together with Pri and Ubr3 (Ray *et al*, 2019). OvoL factors also display evolutionarily conserved role in germ cell precursors (Hayashi *et al*, 2017). In flies, the germline-specific OvoB activator and OvoA repressor are produced from two alternative promoters. OvoB is required for the maintenance of germ cells, while OvoA later acts for their differentiation (Andrews *et al*, 2000; Hayashi *et al*, 2017). Precocious expression of OvoA leads to germ line loss (Andrews *et al*, 2000) and other *ovo* mutations cause ovarian tumors (Oliver *et al*, 1993). Although relying on different mechanisms between soma (post-translational processing) and germline (alternative promoters), the REP-to-ACT switch appears as a key feature of Ovo/Svb function in the control of stem/progenitor cells.

In mammals, OvoLs have been implicated in the reprogramming of mesenchymal fibroblasts toward induced pluripotent stem cells (Kagawa *et al*, 2019) and epithelial lineages (Watanabe *et al*, 2019). OvoLs are also associated with human cancers, in particular those of epithelial origin that often display deregulated Wnt and EGFR signaling (Normanno *et al*, 2006; Zhan *et al*, 2017). Our studies in flies demonstrate opposing effects of Svb[ACT] versus Svb[REP] that promotes or suppresses stem cell-derived tumors, respectively. Interestingly, individual OvoL2 isoforms in mice display strikingly different effects when expressed in patient-derived xenografts, only the OvoL2 repressor can inhibit tumor progression (Watanabe *et al*, 2014). Therefore, OvoL/Svb repressors appear as evolutionarily conserved tumor suppressors, a finding that might open new paths for cancer diagnostic and treatment.

Several studies have shown that OvoL/Svb factors behave as epithelial gatekeepers (Nieto *et al*, 2016), which counteract Snail and Zeb1-2 transcription factors to prevent epithelial to mesenchymal transition (EMT). In agreement with this antagonistic model, *Drosophila* Escargot (Snail) and ZFh1 (Zeb1,2) maintain stemness and prevent ISC differentiation (Korzelius *et al*, 2014; Loza-Coll *et al*, 2014; Antonello *et al*, 2015), while Svb[REP] promotes EC differentiation. However, our results draw a more complex picture, where Svb[ACT] contrariwise cooperate with EMT factors in early stages of the intestinal lineage for the maintenance of ISCs. Indeed, Svb[ACT] can suppress the phenotypes resulting from downregulation of EMT regulators, restoring the pool of stem cells, which display proper cellular architecture. Recent studies show that EMT is not an all-or-none process and instead progresses through a series of reversible intermediate states between the epithelial (E) and mesenchymal (M) phenotypes (Nieto *et al*, 2016). Such hybrid E/M phenotypes are hallmarks of normal and cancer stem cells, and relative doses of EMT factors and OvoL/Svb may provide a tunable window of stemness (Jolly *et al*, 2015).

Taken together, these data show the importance of OvoL/Shavenbaby factors in the control of adult stem cell behavior, in both normal and tumorous conditions. We propose that OvoL/Shavenbaby epithelial factors are ancestral regulators of stemness in animals and their study would provide key insights into stem cell biology. Future work remains to determine how the intrinsic regulatory hub provided by Svb/Pri for intestinal stem cells in flies has evolved both across species and amid the distinct populations of stem cells that regenerate adult organs.

## Materials and Methods

### Reagents and Tools table

| Reagent/Resource | Reference or source | Identifier or catalog number |
|---|---|---|
| **Experimental Models:** *D. melanogaster* | | |
| esg-LacZ: y[1] w[67c23]; P{w[+mC]=lacW}esg[k00606]/CyO | BDSC | BDSC Cat# 10359, RRID:BDSC_10359 |
| Su(H)GBE-LacZ | Furriols and Bray (2001) | N/A |

**Reagents and Tools table**  (continued)

| Reagent/Resource | Reference or source | Identifier or catalog number |
|---|---|---|
| DL-LacZ: ry$^{506}$ P{ry+$^{t7.2}$=PZ}Dl$^{05151}$/ TM3, ry$^{RK}$ Sb$^1$ Ser$^1$ | BDSC | BDSC Cat# 11651, RRID:BDSC_11651 |
| UAS-w-RNAi: y$^1$ v$^1$; P{y$^{+t7.7}$ v$^{+t1.8}$=TRiP.JF01545}attP2 | BDSC | BDSC Cat# 28980RRID:BDSC_28980 |
| UAS-mCherry-RNAi y$^1$ sc* v$^1$ sev$^{21}$; P{y$^{+t7.7}$ v$^{+t1.8}$=VALIUM20-mCherry}attP2 | BDSC | BDSC Cat# 35785 RRID:BDSC_35785 |
| UAS-svb-RNA: i w$^{1118}$; P{GD9026}v41584 | VDRC | Cat# FBst0464178, RRID: FlyBase_FBst0464178 |
| UAS-Ubr3-RNAi: w$^{1118}$; P{GD12698}v22901 | VDRC | Cat# FBst0454736, RRID: FlyBase_FBst0454736 |
| UAS-Notch-RNAi: w*; $^{P\{w+mC}$=UAS-N.dsRNA.P}9G | BDSC | BDSC Cat# 7077, RRID:BDSC_7077 |
| UAS-STAT92E-RNAi: y$^1$ sc* $^{v1}$ sev$^{21}$; P{y$^{+t7.7}$ v$^{+t1.8}$=TRiP.GL00437} attP40/CyO | BDSC | BDSC Cat# 35600, RRID:BDSC_35600 |
| UAS-pri-RNAi: P{UAS-tal.dsRNA} | Galindo et al (2007) | FBtp0072543 |
| UAS-pri: P{UAS-tal}/CyO | Galindo et al (2007) | N/A |
| UAS-EcR-DN: w*; $^{P\{w+mC}$=UAS-EcR.B2.W650A}TP5 | BDSC | BDSC Cat# 9449, RRID:BDSC_9449 |
| UAS-EcR-RNAi$^{#1}$: y$^1$ v$^1$; P{y$^{+t7.7}$ v$^{+t1.8}$=TRiP.HMJ22371}attP40 | BDSC | BDSC Cat# 58286, RRID:BDSC_58286 |
| UAS-EcR-RNAi$^{#2}$: y$^1$ v$^1$; P{y$^{+t7.7}$ v$^{+t1.8}$=TRiP.JF02538}attP2 | BDSC | BDSC Cat# 29374, RRID:BDSC_29374 |
| UAS-Svb$^{ACT(OvoB)}$: w$^{118}$;; P{UAS-ovo.B2} | Payre et al (1999) | FBtp0012383 |
| UAS-Svb$^{REP}$ w$^{118}$;; P{UAS-ovo.svb} | Delon et al (2003) | FBtp0017877 |
| UAS-Svb$^{ACT}$: Y, w;; P{UAS-SvbACT::GFP} | Ray et al (2019) | FBtp0134164 |
| UAS-TCF-DN: y$^1$ w$^{1118}$; P{w$^{+mC}$=UAS-pan.dTCFDeltaN}4 | BDSC | BDSC Cat# 4784, RRID:BDSC_4784 |
| UAS-Arm$^{S10}$: P{w$^{+mC}$=UAS-arm.S10}C, y$^1$ w$^{1118}$ | BDSC | BDSC Cat# 4782, RRID:BDSC_4782 |
| UAS-EGFR-DN: y$^1$ w*; $^{P\{w+mC}$=UAS-Egfr.DN.B}29-77-1; P{w$^{+mC}$=UAS-Egfr.DN.B}29-8-1 | BDSC | BDSC Cat# 5364, RRID:BDSC_5364 |
| UAS-NICD: P{UAS-N.icd} | Cooper and Bray (2000) | FBtp0013654 |
| UAS-DIAP: w*; P{w$^{+mC}$=UAS-DIAP1.H}3 | BDSC | BDSC Cat# 6657 |
| UAS-Wg: w*; P{UAS-wg.h.t:HA1}6C | BDSC | BDSC Cat# 5918, RRID:BDSC_5918 |
| UAS-RasV12: w$^{1118}$; P{w$^{+mC}$=UAS-Ras85D.V12}TL1 | BDSC | BDSC Cat# 4847, RRID:BDSC_4847 |
| esg$^{ts}$: esg-Gal4, UAS-GFP, tubP-Gal80$^{ts}$ | Jiang et al (2009) | N/A |
| NRE$^{ts}$: Su(H)-GBE-Gal4, UAS-GFP; tubP-Gal80$^{ts}$ | Zeng et al (2010) | N/A |
| ISC$^{ts}$: esg-Gal4, UAS-GFP; Su(H)-GBE-GAL80, tubP-Gal80$^{ts}$ | Wang et al (2014) | N/A |
| esg-ReDDM: esg-Gal4, UAS-mCD8::GFP/Cyo; UAS-H2B::RFP, tubP-Gal80$^{ts}$/TM2 | Antonello et al (2015) | N/A |
| MyoIA$^{ts}$: MyoIA-Gal4, UAS-GFP, tubP-Gal80$^{ts}$ | Jiang et al (2009) | N/A |
| Voila$^{ts}$: tubP-Gal80$^{ts}$; Voila-GAL4, UAS-GFP.nls | Balakireva et al (1998) | N/A |
| Act$^{ts}$F/O: hs-FLP; actin < y+< Gal4; UAS–GFP | Chanut-Delalande et al (2014) | N/A |
| esg$^{ts}$F/O: w; esg-Gal4, UAS-GFP, tubP-Gal80$^{ts}$/CyO; UAS-FLP, act > CD2>Gal4/TM6B | Jiang et al (2009) | N/A |
| Pri-Gal4: P{GaWB}tal$^{KG}$/TM3, Sb | Galindo et al (2007) | N/A |
| MARCM-19A: P{ry+$^{t7.2}$=hsFLP}1, P{w$^{+mC}$=tubP-GAL80}LL1 w*, $^{P}$ $_{\{ry+t7.2}$=neoFRT}19A; P{w$^{+mC}$=UAS-mCD8::GFP.L}LL5/ Cyo; P{w$^{+mC}$=tubP-GAL4}LL7/TM6B,Tb | N. Tapon | N/A |

**Reagents and Tools table**   (continued)

| Reagent/Resource | Reference or source | Identifier or catalog number |
|---|---|---|
| svb[R9]: y* w[1118] svb[R9], P{ry[+t7.2]=neoFRT}19A/FM0 | Delon et al (2003) | FBal0151651 |
| Ubr3[B]: y[1] w* Ubr3[B] P{ry[+t7.2]=neoFRT}19A/FM0 | Zanet et al (2015) | FBal0319860 |
| svbE6-lacZ: w[1118]; DmE6-lacZ | Frankel et al (2011) | FBtp0085021 |
| PriA-LacZ | Chanut-Delalande et al (2014) | N/A |
| PriH-LacZ | Chanut-Delalande et al (2014) | N/A |
| PriJ-LacZ | Chanut-Delalande et al (2014) | N/A |
| svbE3N-GFP | This paper | N/A |
| svbE3N-LacZ | Crocker et al (2015) | N/A |
| svbE3N-Pnt-mt-LacZ | This paper | N/A |
| svbE3N-TCF-mt-LacZ | This paper | N/A |
| svb::GFP: E+7-svbP-svb-cDNA::GFP (pRSQ8) | Menoret et al (2013) | N/A |
| E3N-svbP-svb-cDNA | This paper | N/A |
| E3N-TFC-mt-svbP-svb-cDNA | This paper | N/A |
| 9CJ2-LacZ | This paper | N/A |
| 9CJ2-Pdm-mt-LacZ | This paper | N/A |
| **Recombinant DNA** | | |
| Plasmid: placZAttB | DGRC | Cat# 1421 |
| Plasmid: pRSQsvb | Frankel et al (2011) | N/A |
| **Antibodies** | | |
| Mouse monoclonal anti-GFP (1: 500) | Sigma-Aldrich | Cat# 11814460001, RRID:AB_390913 |
| Rabbit anti-GFP (1: 500) | Torrey Pines Biolabs | Cat# TP401 071519, RRID:AB_10013661 |
| Rabbit anti-β-galactosidase (1:500) | MP Biomedicals | Cat# 559761, RRID:AB_2687418 |
| Mouse monoclonal anti-β-galactosidase (1:1,000) | Promega | Cat# 53781, |
| Rat monoclonal antibody to Red Fluorescent Proteins (1:800) | Chromotek GmbH | Cat# 5f8-100, RRID:AB_2336064 |
| Cleaved Drosophila Dcp-1 (Asp216) antibody (1:100) | Cell Signaling Technology | Cat# 9578, RRID:AB_2721060 |
| Mouse anti-Cora antibody (1:100) | DSHB | Cat# C615.16, RRID:AB_1161644 |
| Mouse anti-Prospero antibody (1:100) | DSHB | Cat# MR1A, RRID:AB_528440 |
| Mouse anti-Armadillo (β-catenin) antibody (1:100) | DSHB | Cat# N2 7A1, RRID:AB_528089 |
| Rat anti-DE-Cadherin antibody (1:50) | DSHB | Cat# DCAD2, RRID:AB_528120 |
| Biotinylated goat anti-rabbit IgG antibody (1:1,000) | Vector Laboratories | Cat# BA-1000, RRID: AB_2313606 |
| Rabbit polyclonal anti-Tsp2a antibody (1:1,000) | Izumi et al (2016) | N/A |
| Rabbit polyclonal anti-Scribble antibody (1:1,000) | Chen et al (2018) | N/A |
| Rabbit polyclonal Anti-phospho-Histone H3 (Ser10) (1:1,000) | Millipore | Cat# 06-570, RRID:AB_310177 |
| Sheep Anti-Digoxigenin Fab fragments antibody, Alkaline Phosphatase conjugated (1:2,000) | Roche | Cat# 11093274910, RRID:AB_51449 |
| Goat anti-Rabbit IgG (H+L) Secondary Antibody, AlexaFluor-488 conjugate (1:500) | Thermo Fisher Scientific | Cat# A-11034, RRID:AB_2576217 |
| Goat anti mouse IgG (H+L) secondary antibody, AlexaFluor-488 (1:500) | Quantum Dot Corporation | Cat# 1100-1, RRID:AB_346865 |
| Goat anti-rabbit IgG (H+L) secondary antibody, AlexaFluor-555 (1:500) | Molecular Probes | Cat# A-21428, RRID:AB_141784 |
| Goat anti-mouse IgG (H+L) secondary antibody, AlexaFluor-555 (1:500) | Molecular Probes | Cat# A-21422, RRID:AB_141822 |
| Goat anti-rat IgG (H+L) secondary antibody, AlexaFluor-555 (1:500) | Molecular Probes | Cat# A-21434, RRID:AB_141733 |

**Reagents and Tools table** (continued)

| Reagent/Resource | Reference or source | Identifier or catalog number |
|---|---|---|
| **Oligonucleotides** | | |
| Primer: Fwd_9CJ2: CGGTACCCCGCGGCCGCCATATGTCAACG | This paper | N/A |
| Primer: Rev_9CJ2: TCCGGCGCTCCTCGAGACTATTGGGATACC | This paper | N/A |
| Primer: Fwd_E3-14: CGGTACCCCGCGGCCGCCATATGTCTTTTTTTTTTATCC | This paper | N/A |
| Primer: Rev_E3-14: CCGGCGCTCCTCGAGGTAGGTTAGG | This paper | N/A |
| **Chemicals, enzymes and other reagents** | | |
| Sucrose, BioXtra, >=99.5% (GC) | Sigma-Aldrich | Cat# 57-50-1 |
| DAPI (4′,6-diamidino-2-phenylindole) | Thermo Fisher Scientific | *Cat# D1306* |
| X-Gal (5-bromo-4-chloro-3-indoyl-β-D-Galactopyranoside) | Biosolve | *Cat# 7240-90-6* |
| NBT/BCIP ($C_{40}H_{30}Cl_2N_{10}O_6$ / $C_8H_6NO_4BrClP$ x $C_7H_9N$) | Sigma Aldrich | Cat# 11681451001 |
| 16% Paraformaldehyde, methanol free | Electron microscopy Sciences | Cat# 30525-89-4 |
| Formaldehyde | Electron microscopy Sciences | Cat# 50-00-0 |
| Blocking Reagent for nucleic acid hybridization and detection | Roche | Cat# 11096176001 |
| Phalloidin conjugated to Rhodamin (1:500) | Thermo Fisher Scientific | Cat# R415, RRID:AB_2572408 |
| VECTASHIELD Mounting Medium antibody | Vector Laboratories | Cat# H-1000, RRID:AB_2336789 |
| VECTASHIELD Mounting Medium with DAPI antibody | Vector Laboratories | Cat# H-1200, RRID:AB_2336790 |
| XhoI | New England Biolabs | Cat# R0146L |
| NotI | New England Biolabs | Cat# R0189L |
| Phusion High-Fidelity PCR Master Mix with HF Buffer | Thermo Fisher Scientific | Cat# F531L |
| Phusion High-Fidelity DNA Polymerase | Thermo Fisher Scientific | Cat# F530L |
| **Software** | | |
| ImageJ 1.52a | https://imagej.net/ | RRID:SCR_003070 |
| Fiji | http://fiji.sc | RRID:SCR_002285 |
| Prism 8 | GraphPad | RRID:SCR_002798 |
| Photoshop CC | Adobe | RRID:SCR_014199 |
| FlyBase | http://flybase.org/ | RRID:SCR_006549 |
| Clustal Omega | http://www.ebi.ac.uk/Tools/msa/clustalo/ | RRID:SCR_001591 |
| MUSCLE | http://www.ebi.ac.uk/Tools/msa/muscle/ | RRID:SCR_011812 |
| JASPAR | http://jaspar.genereg.net | RRID:SCR_003030 |
| Clone Manager Software | http://www.scied.com/pr_cmbas.htm | RRID:SCR_014521 |
| ZEN Digital Imaging for Light Microscopy | http://www.zeiss.com/microscopy/en_us/products/microscope-software/zen.html | RRID:SCR_013672 |
| Leica Application Suite | https://www.nikoninstruments.com/Products/Software | RRID:SCR_016555 |
| NIS-Elements | https://www.nikoninstruments.com/Products/Software | RRID:SCR_014329 |
| **Other** | | |
| VECTASTAIN ABC-Peroxidase Kit | Vector Laboratories | Cat# PK-4001, RRID:AB_2336810 |
| Qiaquick PCR Purification kit | Qiagen | Cat# 28104 |
| QIAmp DNA Micro Kit | Qiagen | Cat# 56304 |
| In-Fusion® HD Cloning Plus | Takara | Cat# 638920 |
| DIG RNA Labeling Kit (SP6/T7) | ROCHE | Cat# 11 175 025 910 |

**Methods and Protocols**

### Animal breeding and maintenance

Flies were kept at 25°C and grown on a standard cornmeal food medium (*per* liter: 17 g inactivated yeast powder, 80 g corn flour, 9 g agar, 45 g white sugar, and 17 ml of Moldex). Crosses involving targeted expression under the control of *Gal4/Gal80^ts* were maintained at 18°C until 3–4 days post-hatching, and mated females were shifted to 29°C for 10–14 days for optimal activity of the *UAS/GAL4* system. Flies were transferred to fresh food vials daily. For flip-out (F/O) and MARCM clonal analyses, 3- to 4-day mated adult female flies of the indicated genotypes were heat shocked 1 h at 37°C and then shifted to 25°C for 10 days. The genotype of each *Drosophila* sample is detailed in the Appendix.

### In vivo screening of transcription factors

To avoid indirect effects due to alteration of cell survival/proliferation, the screen was performed in late embryos, when signaling pathways and Svb do not impinge on cell survival and proliferation, as opposed to adult stem cells. Briefly, we knocked down every candidate factor and examined whether it affected the activity of individual *svb* enhancers. We selected transcription factors showing detectable expression in stage-15 whole embryos (Menoret *et al*, 2013) and/or enriched in dorsal trichome cells (Preger-Ben Noon *et al*, 2016), resulting in a list of 227 candidate factors. 273 representative UAS-RNAi lines were obtained from Bloomington and VDRC stock centers, taken from the TRIP or VDRC collection, respectively. Males from each UAS::RNAi carrying line (Table EV1) were crossed with virgin females of stock *w; ptc-Gal4; E3N-lacZ* or *w; ptc-Gal4; 9CJ2-lacZ* and eggs were collected for 12 h at 28°C. Embryos were dechorionated, fixed, and stained using standard protocols (Fernandes *et al*, 2010), with mouse anti-β-galactosidase 1:500 (Promega) and biotinylated goat anti-rabbit (1:1,000) antibodies, revealed using VECTASTAIN ABC Peroxidase Kit (Vector Laboratories). After washing, embryos were mounted in Glycerol/PBS (80/20%) and imaged using a Nikon Eclipse 90i microscope using NIS-elements software (Nikon). Each experiment (typical 200 embryos *per* genotype) was performed at least three times and also included UAS-*w*-RNAi and *w* embryos as negative controls. Reporter patterns upon *RNAi* treatment were classified into "no change", "reduced", or "ectopic" expression; the two latter were kept for additional characterization (Table EV1). For rescuing assays, males carrying *pRSQsvb* constructs were crossed with females of stock *w\* btd^1, svb^1/FM7-kr > GFP,* allowing phenotypical identification of *svb*-mutant embryos. First instar larva cuticles were prepared in Hoyer's/lactic acid 1:1, imaged with phase-contrast microscopy, and trichomes were counted in the ventral region of A6 segments.

### DNA constructs and transgenic lines

DNA fragments from *svb* enhancers was cloned into *placZAttB* for reporter constructs and into *pRSQsvb* for rescue constructs (Frankel *et al*, 2011; Menoret *et al*, 2013; Crocker *et al*, 2015). All plasmids were integrated using the PhiC31 system into the same *attP* landing site (*zh-86F*) by Bestgene (Chino Hills, CA, USA) or in the Payre laboratory. Putative binding sites for transcription factors were identified using JASPAR and their evolutionary conservation was assayed using multiple sequence alignments (Clustal Omega and MUSCLE) of orthologous regions from other species (http://flybase.org). Site-specific mutations were introduced using DNA synthesis

by Genscript, and modified enhancers were sub-cloned in reporter and rescue constructs using ligation-free cloning (In-Fusion, Takara). All constructs were verified by sequencing.

### Immunofluorescence

Stage-15 embryos were processed using standard protocols, using mouse anti-β-galactosidase (1:500, Promega), rabbit anti-Dyl at 1:400 (Fernandes *et al*, 2010), rabbit anti-Min 1:200 (Chanut-Delalande *et al*, 2006) antibodies, Alexa Fluor-488, or Alexa Fluor-555 secondary antibodies at 1:500 (Molecular Probes). Embryos were mounted in Vectashield mounting media (Vector Laboratories) and imaged using X20 and X40 objectives on a Leica Spe confocal laser scanning microscope with Leica Application Suite software, or a Zeiss 710 confocal microscope using the ZEN software (Zeiss).

Adult midguts were dissected in PBS and fixed for 1h at room temperature in a fresh 4% paraformaldehyde solution (Electron microscopy Science) prepared in PBS. Following three washes of 15 min each in 0.1% Triton-PBS (PBST), samples were blocked in 1% BSA-PBST for 30 min at room temperature, prior to overnight incubation with primary antibody at 4°C. The following antibodies were used: mouse and rabbit anti-GFP at 1:500, rabbit anti-β-Galactosidase (MP Biomedicals) at 1:1,000, rat anti-RFP (5F8) at 1:800, cleaved Dcp-1 (Asp216) rabbit antibody at 1:100 (Cell signaling), mouse anti-Prospero (DSHB) at 1:100, rabbit anti-Phospho-Histone 3 (Millipore) at 1:1,000, mouse anti-Coracle at 1:100 (DSHB), mouse anti-β-catenin 1:100 (DSHB), rat anti-DE-Cadherin 1:50 (DSHB), rabbit anti-Pdm1 1:100 (gift from F.J. Díaz-Benjumea), rabbit anti-Tsp2A, and anti-Scribble (gifts from D. St Johnston) at 1:1,000 both. F-Actin was stained with Rhodamine-conjugated phalloidin (Invitrogen, at 1:500). The next day, samples were washed 3 times in PBST for 15 min each and next incubated with Alexa Fluor-488 or Alexa Fluor-555 secondary antibodies at 1:500 (Molecular Probes) for 2 h at room temperature. After three washes, tissues were mounted in Vectashield mounting media containing DAPI (Vector Laboratories) for nuclear staining. Images of posterior midgut were acquired on Leica SPE and SP8 confocal microscopes (X40 objective). 3 to 5 images were acquired for each posterior midgut to cover the 4a to 5 regions.

### In situ hybridization

RNA probes were synthetized from *svb* cDNA (Delon *et al*, 2003) using a DIG RNA Labeling Kit (Roche). Guts were dissected in PBS and fixed in freshly prepared 4% formaldehyde 5 mM EGTA fix solution in PBS. After two washes in PBS, guts were dehydrated in successive baths of methanol (5 times), ethanol (5 times), followed by 1 h in 1:1 xylene/ethanol, and rinsed in methanol. Guts were post-fixed for 30 min at room temperature in the same fix solution and washed in PBS-0.1% Tween 20 (PBSTw). Samples were treated by proteinase-K at room temperature, and the reaction was stopped by a 5-min treatment in 2 mgl/ml glycine, followed by washes in cold PBSTw. Samples were incubated in the fix solution overnight at 4°C and washed in PBST. After 2 hrs. at 60°C in hybridization buffer (HB: 50% formamide, 2X SSC, 1 mg/ml *Tortula* RNA, 0.05 mg/ml Heparin, 2% blocking reagent (Roche), 0.1% CHAPS, 5 mM EDTA, 0.1% Tween 20), guts were incubated overnight with *svb* anti-sense DIG-labeled RNA probe diluted in HB, at 60°C. After several washes in HB, PBSTw, and PBSTw-1% BSA (blocking solution), samples were incubated with anti-DIG primary antibody conjugated to alkaline phosphatase (Roche), at 1:2,000 in blocking solution. After washes in

PBSTw, *in situ* hybridization signals were developed by incubating samples in a fresh staining buffer containing NBT/BCIP stock solution (Sigma Aldrich) diluted at 1:50. Finally, samples were washed and mounted in 50% glycerol/PBS.

### X-Gal staining assays

Adult females were dissected in 1% PBS and guts were fixed in 1% glutaraldehyde-PBS for 15 min at room temperature. Samples were washed three times for 15 min each. The staining buffer (10 mM $Na_2HPO_4$, 1.6 mM $NaH_2PO_4$, 150 mM NaCl, 1 mM $MgCl_2$, 3.5 mM $K_3FeCN_6$, 3.5 mM $K_4FeCN_6$) was warmed up at 37°C for 10 min; an 8% X-Gal solution (5-bromo-4-chloro-3-indoyl-β-D-Galactopyranoside, Sigma Aldrich) was added (final concentration 2.5% X-Gal in SB) and kept for an additional 10 min at 37°C, before centrifugation (5′ at 12,000 *g*). Samples were incubated in staining solution overnight at 37°C, washed three times for 15 min with PBS, and mounted in 50% glycerol/PBS. Bright-field pictures were acquired using a Nikon Eclipse 90i microscope.

### Quantification and statistical analysis

Images were analyzed by using ImageJ, with macros we developed for quantification of indicated cell types (codes are available upon request). Briefly, images were acquired with same setting and transformed into multilayered TIFF files. To count the number of cells positive for a given marker (*e.g.,* GFP), the corresponding channel was used to generate a ROI mask in which DAPI-labeled nuclei were automatically segmented and the number, size, and morphometric features of each object were recorded. Similar analyses were performed for ReDDM assays, quantifying the number of nuclei in $GFP^+/RFP^+$ progenitors versus $GFP^-/RFP^+$ differentiated cells. Data of at least three independent experiments were combined. Statistical analyses were carried out with Prism 8 (GraphPad), using nonparametric unpaired two-tailed Mann–Whitney tests for comparison of two samples, and one-way ANOVA for three or more samples, using Welch's ANOVA with Dunnet's T3 correction for multiple comparisons between samples showing normal distribution (Shapiro–Wilk tests alpha = 0.01) and nonparametric Kruskal–Wallis tests with Dunn's correction for multiple comparison tests when at least one sample did not passed normality test. In each figure, graphs show all individual points, boxes extend from the $25^{th}$ to $75^{th}$ percentiles, whiskers to min and max values, and the horizontal line in each box is plotted at the median. Images were processed and figures drawn using Adobe Photoshop CC.

**Expanded View** for this article is available online.

## Acknowledgements

We would like to thank C. Polesello, all members of the FP laboratory and G. Kolahgar for invaluable help, as well as T. Reiff, S. Ahmed, and B. Edgar for kind sharing of unpublished results. We are also grateful to M. Bou Sleiman and Z. Zhai for discussions and critical reading of the manuscript. We thank P. Valenti for excellent assistance in the early stages of this work, O. Bohner, J. Favier, and A. Destenabes for help in transgenic lines, S. Bosch and the LITC platform (https://www-litc.biotoul.fr/) for imaging. We thank A. Bardin, H. Bellen, D. Bilder, J. Chen, J.P. Couso, A. Debec, F.J. Díaz-Benjumea, M. Dominguez, B. Edgar, N. Frankel, D. Stern, D. St Johnston, and N. Tapon for providing fly stocks and reagents, the Bloomington *Drosophila* Stock Center, Kyoto *Drosophila* Genomics and Genetic Resources, Vienna *Drosophila* Resource Center, Flybase and Developmental Studies Hybridoma Bank for key data and materials. SAH was supported by Fondation pour la Recherche Médicale and Communauté des Communes de Dannieh-Université Libanaise. This work was supported by Agence Universitaire de la Francophonie (PCSI AUF-BMO), École Doctorale des Sciences et de Technologie-Université Libanaise, the Federation of European Biochemical Societies, American University of Beirut URB, Agence Nationale de la Recherche (ANR, ChronoNet), and Fondation pour la Recherche Médicale (DEQ20170336739). The funders had no role in study design, data collection and interpretation, or the decision to submit the work for publication.

## Author contributions

Project setup: DO, SP, and FP; Work conceptualization and supervision: FP and DO; Experiment design: SAH, FP, and DO; Experiments on intestinal stem cells: SAH, DO, JPB, JB, and CI; *in vivo* screening of upstream TFs, transgenic enhancer constructs and genetic rescuing assays: AA; Expertise with microscopy and macros development for automated image analyses: BR; Key resource sharing and materials: ZK and BL; Manuscript writing: SAH, DO and FP. Manuscript edition and comments: All authors.

## Conflict of interest

The authors declare that they have no conflict of interest.

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
