## [Review Process File · The EMBO Journal]

Steroid-dependent switch of OvoL/ Shavenbaby controls self-renewal versus differentiation of intestinal stem cells

Sandy Al Hayek¹, Ahmad Alsawadi, Zakaria Kambris, Jean-Philippe Boquete, Jérôme Bohère, Clément Immarigeon, Brice Ronsin, Serge Plaza, Bruno Lemaitre, François Payre and Dani Osman

DOI: doi.org/10.15252/emj.2019104347

Corresponding authors: Francois PAYRE (francois.payre@univ-tlse3.fr) , Dani OSMAN (dani.osman@ul.edu.lb)

Review Timeline:

Submission Date:	24th Dec 19
Editorial Decision:	11th Feb 20
Revision Received:	9th Oct 20
Editorial Decision:	6th Nov 20
Revision Received:	23rd Nov 20
Accepted:	24th Nov 20

Editor: Daniel Klimmeck

Transaction Report:

Dear Dr Payre,

Thank you for the submission of your manuscript (EMBOJ-2019-104347) to The EMBO Journal. Please accept my sincere apologies for the unusual delay with the peer-review of your manuscript. Your manuscript has been initially sent to three reviewers, however one reviewer got much delayed and in the end did not send us his-her report after repeated chasers. We have received reports from the other two referees, which I enclose below, and decided to proceed with our decision based on these reports.

As you will see, the referees acknowledge the potential interest and novelty of your results, although they also express a number of issues that will have to be conclusively addressed before they can be supportive of publication of your manuscript in The EMBO Journal. In more detail, referee #1 raises concerns that the downstream targets of the ecdysone-pri-Svb axis in the intestinal differentiation context are not sufficiently explored and asks you to address this point (ref#1, pt.1). Further, referee #3 states that the transition between distinct Svb roles during the differentiation cascade should be elaborated on (ref#3, pts.1,2). In addition, the reviewers raise a number of points related to presentation and discussion of the findings as well as wording, which would need to be conclusively addressed to achieve the level of robustness and clarity needed for The EMBO Journal.

I judge the comments of the referees to be generally reasonable and given their overall interest, we are in principle happy to invite you to revise your manuscript experimentally to address the referees' comments.

Please let me know any time if you have additional questions or need further input on the referee comments.

Please see below for additional instructions for preparing your revised manuscript.

Thank you for the opportunity to consider your work for publication. I look forward to your revision.

Kind regards,

Daniel Klimmeck

Daniel Klimmeck, PhD
Editor
The EMBO Journal

When assembling figures, please refer to our figure preparation guideline in order to ensure proper formatting and readability in print as well as on screen:
<http://bit.ly/EMBOPressFigurePreparationGuideline>

Before submitting your revision, primary datasets (and computer code, where appropriate) produced in this study need to be deposited in an appropriate public database (see <https://www.embopress.org/page/journal/14602075/authorguide#datadeposition>).

The accession numbers and database should be listed in a formal "Data Availability" section (placed after Materials & Method) that follows the model below (see also <https://www.embopress.org/page/journal/14602075/authorguide#availabilityofpublishedmaterial>). Please note that the Data Availability Section is restricted to new primary data that are part of this study.

Data availability

Our journal also encourages inclusion of *data citations in the reference list* to directly cite datasets that were re-used and obtained from public databases. Data citations in the article text are distinct from normal bibliographical citations and should directly link to the database records from which the data can be accessed. In the main text, data citations are formatted as follows: "Data ref: Smith et al, 2001" or "Data ref: NCBI Sequence Read Archive PRJNA342805, 2017". In the Reference list, data citations must be labeled with "[DATASET]". A data reference must provide the database name, accession number/identifiers and a resolvable link to the landing page from which the data can be accessed at the end of the reference. Further instructions are available at <https://www.embopress.org/page/journal/14602075/authorguide#referencesformat>

- a point-by-point response to the referees' comments, with a detailed description of the changes made (as a word file).
- a word file of the manuscript text.
- individual production quality figure files (one file per figure)

- a complete author checklist, which you can download from our author guidelines (<http://emboj.embopress.org/authorguide>).

- Expanded View files (replacing Supplementary Information)

Further information is available in our Guide For Authors:

The revision must be submitted online within 90 days; please click on the link below to submit the revision online before 11th May 2020.

Link Not Available

Referee #1:

In this manuscript the authors analyzed Svb function in intestinal stem cells. Svb was processed by Pri to be Svb activator, which maintained ISC. Svb expression was regulated by Wnt and EGFR pathway through TCF and Pnt, respectively. On the other hand, the unprocessed repressor Svb promoted differentiation in enterocytes. The expression of Svb in this stage is regulated by Pou factor Pdm1. The authors further showed that Svb repressor acted as a tumor suppressor. Although the focus of this study is obscure, the finding that switching the two opposing functions of a transcription factor regulate the fate of stem cells as well as of cancer is interesting. However, this manuscript needs to address the points raised below.

Major issue

Although there is much data about phenotypes of Svb mutants, expression analysis of target genes is insufficient. In particular, as OvoL factors is known to be a strong regulator of EMT (MET), analysis of EMT-related genes should be included.

Minor points

1. In Introduction, "OvoL factors are required for the differentiation of neural progenitors..." is misleading. Contrary, OvoL factors suppress differentiation of neuroectoderm lineage (Mackay et al., Dev. Biol. 291:38, 2006; Zhang et al., J. Biol. Chem. 288:6166, 2013).
2. This is an additional non-essential suggestion; Some sentences of this manuscript (meaning or claims) are difficult to understand, so I recommend proofreading by colleagues or English editing.

Referee #3:

This is an interesting manuscript that puts forward the idea that smORF-dependent processing of the Shavenbaby (Svb) transcription factor, a member of the OvoL family of transcription factors implicated in epithelial tumours, acts as a switch to regulate the transition between intestinal stem cell proliferation and differentiation; in stem cells the Pri smORF peptide contributes to svp processing to yield an activator version of Svp that sustains stem cell identity. In enterocytes (the stem cell progeny), the lack of Pri peptides results in an Svp repressor form, which allows enterocyte differentiation. The authors also report that steroid (ecdysone) signalling promotes expression of Pri peptides in stem cells, potentially providing a link between this hormone and stem cell proliferation.

The data is generally of high quality and the authors' conclusions are typically based on several independent experiments. There are only a few inconsistencies/premature conclusions that need to be directly addressed.

1. The authors clearly show in different ways that svb is required to keep stem cells alive. The phenotypic analysis of pri and Ecdysone receptor (EcR) loss of function is more limited and does not allow them to distinguish between their effects on stem cell survival and proliferation. This is an important point because, if ecdysone is upstream of pri which in turn controls the Svp repressor to activator switch, ecdysone knockdown should kill stem cells (like svp loss-of-function), so the rescue shown in Fig 2E/S2D should not happen. To make matters more complicated, a previous study that made mutant EcR clones failed to observe effects on stem cell survival/proliferation (see Fig S3 in doi:10.1242/dev.083030). EcR and pri mutant clones should allow the authors to draw firmer conclusions.

2. My interpretation of the data and model that the authors present is that pri expression is switched off somewhere in the stem cell to enterocyte transition, possibly in an ecdysone independent way. Would the authors agree with this? If so, simultaneously blocking ecdysone receptor and cell death in progenitors should not impact enterocyte svp expression or fate. Is that the case? Related to this, how is svp expression induced in ECs? Is it maintained from the progenitors (in which case it should also be TCF/Pnt dependent) or is there an EC-specific activation of svp?

3. For readers who are unfamiliar with the ovo/svb locus, it would be very helpful to have a visual representation of the different isoforms/enhancers/expression, as well as the locations of the reporters used in this study and what isoforms they are likely to report on. Could Figure 1 be expanded (or include a new supplementary panel)? Related to this, is the ovoD1 allele (commonly used to create sterile flies) expected to affect gut-specific isoforms?

We thank both reviewers for insightful comments and appreciate their constructive suggestions to improve the manuscript.

Every point has faithfully been taken into account and the revised manuscript has been extensively edited. The main text was carefully checked and several parts were rewritten. We also redrew most figures and extended views to clarify presentation and incorporate new data. Besides schematic representation of the *ovo/svb* locus and protein isoforms asked by a reviewer (**Fig 1 A**), other drawings have been modified or introduced to improve readability (**Figs 1B, 2C, 4C, 5A, 5C, 8D**).

Additional fresh results are shown in:

Fig 3C and D: analysis of EMT-related factors and functional interaction between SvbACT and EMT-inducing factors (Esg, Zfh1, miR8).

Fig 5G: further characterization of Svb^{REP} effects when driven in stem/progenitor cells.

Fig 8A, B, C: modification of enterocyte epithelial organization, increased mitosis and dysplasia resulting from ectopic maturation of Svb in ECs.

Fig EV3A: stem/progenitor hyperplasia resulting from elevated Svb^{ACT} levels.

Fig EV3C: ISC/EB-specific expression of a Svb target enhancer, *snE1*, and abrogation of its activity upon knocking out Svb binding sites.

Referee #1:

In this manuscript the authors analyzed Svb function in intestinal stem cells. Svb was processed by Pri to be Svb activator, which maintained ISC. Svb expression was regulated by Wnt and EGFR pathway through TCF and Pnt, respectively. On the other hand, the unprocessed repressor Svb promoted differentiation in enterocytes. The expression of Svb in this stage is regulated by Pou factor Pdm1. The authors further showed that Svb repressor acted as a tumor suppressor. Although the focus of this study is obscure, the finding that switching the two opposing functions of a transcription factor regulate the fate of stem cells as well as of cancer is interesting. However, this manuscript needs to address the points raised below.

Major issue

R1.1 Although there is much data about phenotypes of Svb mutants, expression analysis of target genes is insufficient.

The revised version provides new data to address this point, through examination of Fascin, a known epithelial direct target of Svb (Chanut *et al.*, PLoS Biol. 2006). These experiments show that a Fascin enhancer activated by Svb^{ACT} (Menoret *et al.*, Genome Biol. 2013) is specifically expressed in ISC/EBs, and that mutations of Svb binding sites abrogate its activity in the adult midgut (**Fig EV3C**).

In particular, as OvoL factors is known to be a strong regulator of EMT(MET), analysis of EMT-related genes should be included.

We agree with the reviewer and performed accordingly a new series of experiments to address this point.

Specifically, we analyzed the distribution of key EMT-related factors (according to recent guidelines for research on EMT, see Yang *et al.*, Nature Rev Mol Cell Biol, 2020), including E-cadherin and β -catenin that feature epithelial cell junctions, and Scribble, which determines more lateral domains. These data show that Svb^{ACT} triggers the acquisition of stem-cell like epithelial organization, both in supernumerary precursors (**Fig 3C**), or following Svb^{ACT} ectopic expression in ECs (**Fig 8 A,A'**).

We also analyzed functional interactions between Svb and EMT-inducing factors Esg (Snail) and Zfh1 (Zeb), as well as *miR8* (*miR200*). The upregulation of *miR8* represses Esg and Zfh1 levels and thus leads to the loss of ISC/EBs, which prematurely differentiate (Antonello *et al.*, EMBO J. 2015). Our results show that Svb^{ACT} is sufficient to rescue this phenotype, restoring stem/precursors cells that display proper distribution of EMT-related factors (**Fig 3D**).

We believe that the reviewer's suggestion has genuinely improved our manuscript and these results provide additional insights into the role of Svb^{ACT} in the control of epithelial stem cells.

Minor points

R1.2. In Introduction, "OvoL factors are required for the differentiation of neural progenitors..." is misleading. Contrary, OvoL factors suppress differentiation of neuroectoderm lineage (Mackay et al., Dev. Biol. 291:38, 2006; Zhang et al., J. Biol. Chem. 288:6166, 2013).

The sentence has been modified according to the reviewer's suggestion:
"human OvoL2 is required for the maintenance of corneal epithelium cells (Kitazawa et al., 2016) and its alteration is a major cause of inherited corneal dystrophies (Davidson et al., 2016)."

R1.3. This is an additional non-essential suggestion; Some sentences of this manuscript (meaning or claims) are difficult to understand, so I recommend proofreading by colleagues or English editing.

The manuscript has been edited in depth, several sections have been entirely rewritten, and we thank English-speaker colleagues for their help.
We hope that this work will help clarifying our message.

Referee #3:

This is an interesting manuscript that puts forward the idea that smORF-dependent processing of the Shavenbaby (Svb) transcription factor, a member of the OvoL family of transcription factors implicated in epithelial tumours, acts as a switch to regulate the transition between intestinal stem cell proliferation and differentiation; in stem cells the Pri smORF peptide contributes to Svb processing to yield an activator version of Svb that sustains stem cell identity. In enterocytes (the stem cell progeny), the lack of Pri peptides results in an Svb repressor form, which allows enterocyte differentiation. The authors also report that steroid (ecdysone) signalling promotes expression of Pri peptides in stem cells, potentially providing a link between this hormone and stem cell proliferation.

The data is generally of high quality and the authors' conclusions are typically based on several independent experiments. There are only a few inconsistencies/premature conclusions that need to be directly addressed.

R3.1. The authors clearly show in different ways that Svb is required to keep stem cells alive. The phenotypic analysis of *pri* and Ecdysone receptor (EcR) loss of function is more limited and does not allow them to distinguish between their effects on stem cell survival and proliferation. This is an important point because, if ecdysone is upstream of *pri* which in turn controls the Svb repressor to activator switch, ecdysone knockdown should kill stem cells (like *svb* loss-of-function), so the rescue shown in Fig 2E/S2D should not happen.

As detailed below, two papers published this summer nicely demonstrate that EcR is required for stem cell proliferation (Ahmed *et al.*, Nature 2020; Zipper *et al.*, ELife 2020). This brings additional support to our model in which EcR downregulation should decrease *pri* expression and, thereby, should reduce Svb maturation from Svb^{REP} to Svb^{ACT}. EcR^{DN} is thus not expected to mimic the loss of *svb* function (cell death), but the accumulation of unprocessed Svb^{REP} in stem/precursor cells, which leads to their premature differentiation. Therefore, expression of *pri* or Svb^{ACT} should rescue the EcR^{DN} phenotypes, as observed in both conditions.

To make matters more complicated, a previous study that made mutant EcR clones failed to observe effects on stem cell survival/proliferation (see Fig S3 in [doi:10.1242/dev.083030](https://doi.org/10.1242/dev.083030)). *EcR* and *pri* mutant clones should allow the authors to draw firmer conclusions.

The reviewer was right, a putative function of ecdysone in adults intestinal stem cells has remained a debated issue for a while. The aforementioned studies now provide compelling evidence of the role of EcR, its cofactor Usp, and the downstream nuclear receptor Eip75B/PPAR γ , as well as the transcription factors Broad and Hr3, in promoting stem cell proliferation (Ahmed *et al.*, Nature 2020; Zipper *et al.*, ELife 2020).

Interestingly, high levels of ecdysone are produced from ovaries as a post-mating response, while virgin females (or males) have far weaker ecdysone titers, which may explain why some colleagues failed to detect ecdysone function in ISCs. For all of our experiments, we used 3 days-old mated females, as now explicitly stated in the main text.

In any case, these independent studies fully support our conclusions, further suggesting that upregulation of *pri* expression is an important target of ecdysone signaling to promote proliferation of stem/precursor cells in the midgut.

Concerning *pri* mutant clones, their interpretation is complicated by long-distance non-cell autonomous effects (Galindo *et al.*, PLoS Biol. 2007; Kondo *et al.*, Nat Cell Biol. 2007, Kondo *et al.*, Science 2010; Chanut *et al.*, Nat Cell Biol 2014), likely resulting from diffusion of these tiny peptides (<1,5kDa) across epithelial tissues. In our hands, *pri* clones in the eye need to be larger than one third/half of the full tissue to reveal a phenotype. In addition, *pri* clones are slow growing and required to be placed in a *minute*/+ background, to avoid being eliminated by cell competition. These conditions are particularly challenging when dealing with intestinal stem cells, which are very sensitive to culture conditions, food, and genetic background.

This is actually why we used mosaic mutant clones for Ubr3, the direct target effector of Pri peptides. Ubr3 is the E3 ligase that directly binds to Svb, triggers Svb polyubiquitination and targeting to proteasome. Unlike *pri*, Ubr3 displays cell-autonomous function and doesn't require *minute* background. From our accumulated experience, *Ubr3* clones provide the ultimate and "cleanest" way to abrogate Svb processing.

R3.2. My interpretation of the data and model that the authors present is that *pri* expression is switched off somewhere in the stem cell to enterocyte transition, possibly in an ecdysone independent way. Would the authors agree with this?

Yes, this is correct. Consistent with this interpretation, we found that *pri* expression is normally specific to ISC/EBs (**Fig 2B**), and the ectopic expression of *pri* in ECs is sufficient to cause phenotypes reminiscent to those observed upon expression of constitutive Svb^{ACT} (**Fig 7D**).

If so, simultaneously blocking ecdysone receptor and cell death in progenitors should not impact enterocyte Svb expression or fate. Is that the case?

Ecdysone signaling is not expected to impinge upon *svb* transcription, as previously demonstrated during development (Chanut *et al.*, Nat Cell Biol 2014).

Concerning the fate of Svb, as also discussed above (R3.1), our model predicts that EcR knockdown downregulates *pri* expression leading, in turn, to accumulation of unprocessed Svb repressor. Consistently, co-expression of *pri* with EcR^{DN} suppresses ISC/EB loss and, similarly, expression of constitutive Svb activator rescues EcR^{DN} phenotypes (**Figs 2F and EV2C**). Conversely, high Svb^{REP} levels in ISC/EBs trigger differentiation (**Figs 5D,E,FG and 6C**), unless *pri* levels are simultaneously increased (**Fig EV3A,B**).

That excessive Svb^{REP} does not lead to cell death and imposes differentiation into ECs is also well illustrated by prominent increase in cell size and nuclei ploidy (**Fig 5D**), lack of apoptotic staining and absence of rescue by DIAP1 (**Fig 5E,E'**), and firmly demonstrated by clonal analysis and lineage tracing (**Fig 5E,F**).

The active role of Svb^{REP} in promoting differentiation, which is epistatic to each of the pathways we have tested, is also manifest in hyperplastic/tumorous contexts (**Fig 6C**).

Finally, the importance of Svb^{REP} in promoting and maintaining EC differentiation is reinforced by new data. They show that ectopic maturation of Svb into the activator form disrupts epithelium organization, imposes redistribution of EMT-related factors, and promotes cell division and dysplasia (**Fig 8**).

Corresponding sections of the manuscript have been extensively resented in order to clarify this important point.

Related to this, how is *svb* expression induced in ECs? Is it maintained from the progenitors (in which case it should also be TCF/Pnt dependent) or is there an EC-specific activation of *svb*?

Indeed, our results indicate that a distinct enhancer driving *svb* expression, called *9CJ2*, is only active in ECs. We show that *9CJ2* function relies on Pdm1 (**Fig 5C**), a POU/homeodomain transcription factor that is a hallmark of differentiated ECs (Jiang *et al.*, Cell 2009; Beebe *et al.*, Dev Biol. 2010).

As suggested in our model, *svb* expression throughout the intestinal stem cell lineage involves two separate sets of regulatory interactions between *svb* enhancers and upstream transcription factors, *i.e.*:

- i) the E3N *svb* enhancer, activated by Pnt and TCF in ISC/EBs (**Fig 4C**),
- ii) then the *9CJ2* *svb* enhancer, activated by Pdm1 in ECs (**Fig 5C**).

Therefore, E3N enhancer initiates *svb* expression in ISC/EBs while *9CJ2* enhancer maintains *svb* expression in ECs. Obviously, this doesn't rule out some transient overlap when EBs progressively differentiate into ECs.

R3.3. For readers who are unfamiliar with the *ovo/svb* locus, it would be very helpful to have a visual representation of the different isoforms/enhancers/expression, as well as the locations of the reporters used in this study and what isoforms they are likely to report on. Could Figure 1 be expanded (or include a new supplementary panel)? Related to this, is the *ovoD1* allele (commonly used to create sterile flies) expected to affect gut-specific isoforms?

We thank the reviewer for this suggestion and accordingly we modified the **Figure 1** to provide relevant information.

It is well established that the *Ovo^{D1}* mutation has no detectable effect in somatic tissues (this is actually why *Ovo^{D1}* animals are viable) and doesn't affect the production of Shavenbaby proteins. Consistently, Zipper and colleagues also confirm that *Ovo^{D1}* females don't display altered ISC proliferation (Zipper *et al.*, ELife 2020).

Dear François,

Thank you for submitting your revised manuscript (EMBOJ-2020-104729R) to The EMBO Journal. My apologies for getting back to you with delay due to protracted reviewer input. Your amended study was sent back to the two referees for re-evaluation, however referee #1 was at this time not able to reassess your work. Please note that we have editorially assessed your response to the critique raised and found it to be satisfactorily addressed. We have received comments from referee #3, which I enclose below. As you will see this referee finds that his-her concerns have been sufficiently addressed and is now broadly in favour of publication.

Thus, we are pleased to inform you that your manuscript has been accepted in principle for publication in The EMBO Journal, pending the minor remaining issues related to formatting and data representation as detailed below are addressed at re-submission.

Please contact me at any time if you have further questions related to below points.

As you may have seen, every paper now includes a 'Synopsis', displayed on the html and freely accessible to all readers. The synopsis includes a 'model' figure as well as 2-5 one-short-sentence bullet points that summarize the article. I would appreciate if you could provide this figure and the bullet points.

Thank you for giving us the chance to consider your manuscript for The EMBO Journal. I look forward to your final revision.

Again, please contact me at any time if you need any help or have further questions.

Kind regards,

Daniel

Daniel Klimmeck PhD
Editor
The EMBO Journal

>> Introduce ORCID IDs for all corresponding authors (D.O.) via our online manuscript system. Please see below for additional information.

>> Rename the current 'Declaration of Interests' section to 'Conflict of Interest'.

>> Please add zoom boxes in Fig 5D.

>> Dataset EV legends: The excel table labelled " Tables S1" should be uploaded as "Table EV1",

file type dataset. Legends in the file and callouts need to be adjusted accordingly.

>>Please turn the current source data table on genotypes into an 'Appendix' .pdf file, including a ToC on its first page, and adjust callouts in the main text and legends accordingly.

>> Please consider additional changes and comments from our production team as indicated by the .doc file enclosed and leave changes in track mode.

Please note that as of January 2016, our new EMBO Press policy asks for corresponding authors to link to their ORCID iDs. You can read about the change under "Authorship Guidelines" in the Guide to Authors here: <http://emboj.embopress.org/authorguide>

In order to link your ORCID iD to your account in our manuscript tracking system, please do the following:

1. Click the 'Modify Profile' link at the bottom of your homepage in our system.
2. On the next page you will see a box half-way down the page titled ORCID*. Below this box is red text reading 'To Register/Link to ORCID, click here'. Please follow that link: you will be taken to ORCID where you can log in to your account (or create an account if you don't have one)
3. You will then be asked to authorise Wiley to access your ORCID information. Once you have approved the linking, you will be brought back to our manuscript system.

We regret that we cannot do this linking on your behalf for security reasons. We also cannot add your ORCID iD number manually to our system because there is no way for us to authenticate this iD number with ORCID.

Thank you very much in advance.

- a point-by-point response to the referees' comments, with a detailed description of the changes made (as a word file).
- a word file of the manuscript text.

- individual production quality figure files (one file per figure)
 - a complete author checklist, which you can download from our author guidelines (<https://www.embopress.org/page/journal/14602075/authorguide>).
 - Expanded View files (replacing Supplementary Information)
- Please see out instructions to authors
<https://www.embopress.org/page/journal/14602075/authorguide#expandedview>

The revision must be submitted online within 90 days; please click on the link below to submit the revision online before 4th Feb 2021.

Referee #3:

The authors have addressed my concerns. Their data adds to two recently published papers on ecdysone and ISC proliferation by shedding mechanistic light on the molecular processes downstream of ecdysone.

The authors performed the requested changes.

Dear Dr Payre, dear Dr Osman,

Thank you for submitting the revised version of your manuscript. I have now evaluated your amended manuscript and concluded that the remaining minor concerns have been sufficiently addressed.

Thus, I am pleased to inform you that your manuscript has been accepted for publication in the EMBO Journal.

Please note that it is EMBO Journal policy for the transcript of the editorial process (containing referee reports and your response letter) to be published as an online supplement to each paper.

Also in case you might NOT want the transparent process file published at all, you will also need to inform us via email immediately. More information is available here:

http://emboj.embopress.org/about#Transparent_Process

Please note that in order to be able to start the production process, our publisher will need and contact you regarding the following forms:

- PAGE CHARGE AUTHORISATION (For Articles and Resources)

[http://onlinelibrary.wiley.com/journal/10.1002/\(ISSN\)1460-2075/homepage/tej_apc.pdf](http://onlinelibrary.wiley.com/journal/10.1002/(ISSN)1460-2075/homepage/tej_apc.pdf)

- LICENCE TO PUBLISH (for non-Open Access)

Your article cannot be published until the publisher has received the appropriate signed license agreement. Once your article has been received by Wiley for production you will receive an email from Wiley's Author Services system, which will ask you to log in and will present them with the appropriate license for completion.

- LICENCE TO PUBLISH for OPEN ACCESS papers

Authors of accepted peer-reviewed original research articles may choose to pay a fee in order for their published article to be made freely accessible to all online immediately upon publication. The EMBO Open fee is fixed at \$5,200 (+ VAT where applicable).

We offer two licenses for Open Access papers, CC-BY and CC-BY-NC-ND.

For more information on these licenses, please visit: <http://creativecommons.org/licenses/by/3.0/> and http://creativecommons.org/licenses/by-nc-nd/3.0/deed.en_US

- PAYMENT FOR OPEN ACCESS papers

You also need to complete our payment system for Open Access articles. Please follow this link and select EMBO Journal from the drop down list and then complete the payment process:

https://authorservices.wiley.com/bauthor/onlineopen_order.asp

Should you be planning a Press Release on your article, please get in contact with embojournal@wiley.com as early as possible, in order to coordinate publication and release dates.

On a different note, I would like to alert you that EMBO Press is currently developing a new format for a video-synopsis of work published with us, which essentially is a short, author-generated film explaining the core findings in hand drawings, and, as we believe, can be very useful to increase visibility of the work.

Please see the following link for a representative example:

The video-synopses are embedded in the respective article html page, see e.g.

<https://www.embopress.org/doi/full/10.15252/embj.2019103932>

Finally, we have noted that the submitted version of your article is also posted on the preprint platform bioRxiv. We thus appreciate if you could alert bioRxiv on the acceptance of this manuscript at The EMBO Journal in order to allow for an update of the entry status. Thank you in advance!

If you have any questions, please do not hesitate to call or email the Editorial Office.

Kind regards,

Daniel Klimmeck

Daniel Klimmeck, PhD
Editor
The EMBO Journal
EMBO
Postfach 1022-40
Meyershofstrasse 1
D-69117 Heidelberg
contact@embojournal.org
Submit at: <http://emboj.msubmit.net>

Corresponding Author Name: François Payre and Dani Osman

Journal Submitted to: The EMBO journal

Manuscript Number: EMBOJ-2019-104347